# BRiTE: Bootstrapping Reinforced Thinking Process to Enhance Language Model Reasoning

**Han Zhong** [*1] **Yutong Yin** [*2] **Shenao Zhang** [*2] **Xiaojun Xu** [*3] **Yuanxin Liu** [*2] **Yifei Zuo** [*2] **Zhihan Liu** [*2]
**Boyi Liu** [3] **Sirui Zheng** [2] **Hongyi Guo** [2] **Liwei Wang** [1] **Mingyi Hong** [4] **Zhaoran Wang** [*2]

## Abstract

Large Language Models (LLMs) have demonstrated remarkable capabilities in complex reasoning tasks, yet generating reliable reasoning processes remains a significant challenge. We present a unified probabilistic framework that formalizes LLM reasoning through a novel graphical model incorporating latent thinking processes and evaluation signals. Within this framework, we introduce the Bootstrapping Reinforced Thinking Process (BRiTE) algorithm, which works in two steps. First, it generates high-quality rationales by approximating the optimal thinking process through reinforcement learning, using a novel reward shaping mechanism. Second, it enhances the base LLM by maximizing the joint probability of rationale generation with respect to the model's parameters. Theoretically, we demonstrate BRiTE's convergence at a rate of $1/T$ with $T$ representing the number of iterations. Empirical evaluations on math and coding benchmarks demonstrate that our approach consistently improves performance across different base models without requiring human-annotated thinking processes. In addition, BRiTE demonstrates superior performance compared to existing algorithms that bootstrap thinking processes use alternative methods such as rejection sampling, and can even match or exceed the results achieved through supervised fine-tuning with human-annotated data.

---
*These authors are the main contributors to this work; detailed contributions are provided in Appendix A. [1]Center for Dada Science, Peking University [2]Northwestern University [3]Bytedance Research [4]University of Minnesota. Correspondence to: Han Zhong <hanzhong@stu.pku.edu.cn>.

*Proceedings of the 42$^{nd}$ International Conference on Machine Learning*, Vancouver, Canada. PMLR 267, 2025. Copyright 2025 by the author(s).

## 1. Introduction

Large Language Models (LLMs; OpenAI, 2023; Anthropic, 2023; Team et al., 2024a), have emerged as a breakthrough in artificial intelligence, demonstrating unprecedented capabilities in natural language processing and generation. The training pipeline of these state-of-the-art models consists of two critical phases: pre-training and post-training. During the *pre-training* phase, LLMs learn from vast datasets to predict subsequent tokens in sequences, enabling them to learn extensive linguistic patterns, contextual understanding, and general world knowledge. The *post-training* phase further refines these models through two stages: Supervised Fine-tuning (SFT) and Reinforcement Learning from Human Feedback (RLHF; Christiano et al., 2017; Ziegler et al., 2019; Ouyang et al., 2022). Recent research (OpenAI, 2024) has shown that by scaling the inference time, these models demonstrate sophisticated *reasoning* capabilities, particularly in domains such as mathematics and programming.

Unlocking LLM reasoning abilities typically relies on structured prompting methods that break down problems into step-by-step solutions, known as chain-of-thought (CoT) reasoning (Wei et al., 2022). While this approach has shown promise and inspired various extensions (Wang et al., 2022; Yao et al., 2024), the fundamental challenge of reasoning reliability remains unresolved. Generated rationales often lack logical completeness or validity, with their quality heavily dependent on task-specific prompting strategies. Recent developments in inference-time scaling techniques (Snell et al., 2024) have shown potential improvements. However, these approaches primarily address surface-level symptoms rather than the core challenge of generating high-quality reasoning processes. Furthermore, the field increasingly seeks automated improvements to reduce reliance on manual prompt engineering. This context motivates our designing mechanism for high-quality thinking process generation. Meanwhile, prior research (e.g., Zelikman et al., 2022; Yuan et al., 2023) indicates that reasoning processes, when properly selected via verifiers, can enhance an LLM's reasoning capabilities during post-training. This leads to our research objective: *developing a framework for the automated generation of high-quality (correct) reasoning processes and*

*incorporating them into the post-training stage to improve existing algorithms.*

To this end, we propose a unified probabilistic framework that formalizes the reasoning process accompanying evaluation signals, followed by a generic algorithmic framework. Specifically, our work has three key contributions:

- We formulate the problem as a probabilistic graphical model (Figure 1), characterizing the generation flow from prompt $X$ to latent rationales $Z$ to answer $Y$, along with their corresponding evaluation signal $O$. This explicit mathematical characterization serves two essential purposes: first, introducing $Z$ breaks down the complex distribution $\mathbb{P}(Y \mid X)$ into more tractable marginal distributions $\mathbb{P}(Z \mid X)$ and $\mathbb{P}(Y \mid X, Z)$, which aligns with Chain-of-Thought (CoT) methods (Wei et al., 2022); second, introducing $O$ provides crucial rationale-answer quality signal, making the generation of desired (correct) rationales more achievable.

- Under this framework, our learning objective is to maximize the probability of generating high-quality rationales and answers that yield optimal evaluation signals. To achieve this, we propose the Bootstrapping Reinforced Thinking Process (BRiTE) algorithm consisting of two stages: first generating high-quality rationales by training an LLM whose output approximates the desired posterior of thought given the question-answer pair; and then fine-tuning the seed LLM by maximizing the joint probability of rationale generation with respect to the LLM's parameters. Theoretically, we prove our algorithm converges at a rate of $1/T$, where $T$ is the number of iterations. Regarding the practical implementation, to address the challenging Bayesian inference problem in the former step, we develop a novel reward shaping mechanism that converts it into a reinforcement learning optimization problem.

- Our empirical evaluations of math and code generation benchmarks show consistent improvements across multiple LLM series, including Gemma, Llama, and Mistral. Our experimental results demonstrate BRiTE's ability to enhance existing post-training algorithms (rejection sampling type methods and iterative DPO) through RL-based rationale generation with consistent improvements. Notably, BRiTE achieves a 10-point improvement on GSM8K benchmarks when applied to the Gemma-1.1-7B-it base model. Notably, our algorithm matches or even exceeds the performance of supervised fine-tuning methods that use human-labeled thinking processes, despite not requiring any human-annotated data.

In summary, we present a provable and practical framework for automated thinking process generation that can be seamlessly integrated into the training stage. The thinking

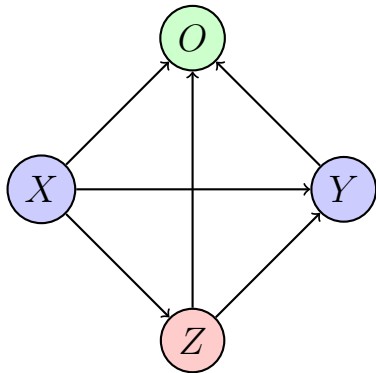

*Figure 1.* LLM as a probabilistic graphical model. $X$ and $Y$ represent prompt and response, respectively. The latent variable $Z$ indicates the intrinsic thinking process behind generation. Evaluation signal $O$ is influenced by $X$, $Z$, and $Y$.

processes generated by our framework are of high quality, surpassing not only those produced through CoT prompting, but also outperforming human-annotated thinking processes when applied to fine-tuning. Our framework represents a significant advancement in improving LLM reasoning capacity through the creation of synthetic data that incorporates detailed thinking processes (CoT data).

### 1.1. Related Works

We discuss mostly related works here. More discussion on RLHF is deferred to Appendix B.

**Reasoning in LLMs.** Prior work has explored various prompting techniques to enhance language model reasoning capabilities. CoT prompting (Wei et al., 2022) has emerged as a particularly effective approach, encouraging models to break down complex problems into intermediate steps by demonstrating step-by-step reasoning paths. The following works, such as Zhou et al. (2022); Yao et al. (2024), design more prompting techniques to enhance the model's capacity. However, these methods typically rely on manually crafted (CoT) prompts to elicit reasoning processes, which are then used to guide the model's generation process. While such approaches have shown promising results in improving model performance across various reasoning tasks, they remain dependent on human-designed prompting templates and may not fully capture the natural reasoning patterns that emerge during model inference. To this end, a line of works aims to boost the latent reasoning process quality or even achieve automatic reasoning process generation, where the latter one is our focus but our method is based on reinforcement learning and thus is different from previous works. Previous methods can be roughly regarded as an EM-type algorithm, and detailed comparisons are presented below.

**EM-type Methods.** Our algorithmic framework builds upon the Expectation-Maximization (EM) algorithm (Dempster et al., 1977) and its variant, rejection sampling EM (Neal

& Hinton, 1998; Rush & Ritter, 2024). In standard EM, the E-step learns a posterior over latent variables while the M-step maximizes the expected log-likelihood; rejection sampling EM approximates complex posteriors through rejection sampling in the E-step before proceeding with the standard M-step. This motivates various rejection sampling fine-tuning methods employed in modern LLM training (e.g., Cobbe et al., 2021; Zelikman et al., 2022; Dong et al., 2023a; Yuan et al., 2023; Gulcehre et al., 2023; Singh et al., 2023; Zelikman et al., 2024; Yuan et al., 2024; Chen et al., 2024). These algorithms filter outputs or/and latent reasoning process through reward functions or verifiers (E-step), then perform fine-tuning on the selected samples (M-step). Our algorithm generalizes these approaches (Example 3.6) and introduces a novel E-step based on reinforcement learning (Section 3.4). While Hu et al. (2023); Hoffman et al. (2024) also conceptualize LLM generation as latent variable models and propose EM-type algorithms based on Markov chain Monte Carlo (MCMC) or generative flow networks, our work distinctively focuses on enhancing the reasoning capabilities of LLMs through automated reasoning processes using reinforcement learning (e.g., PPO). Notably, in contrast to these prior works, we provide a more general and rigorous mathematical framework fro LLM reason and unified theoretical guarantees. Furthermore, our framework extends beyond supervised and rejection sampling fine-tuning to advance iterative direct preference learning (Xiong et al., 2024a), contributing to improved LLM reasoning in the RLHF paradigm. Finally, two recent works by Liu et al. (2024a) and Wang et al. (2024) also apply RL techniques to enhance LLM reasoning. However, these works do not focus on improving the thinking process generation, making them not directly comparable to our approach.

### 1.2. Notations

For any space $\mathcal{X}$, we denote $\Delta(\mathcal{X})$ as the set of distributions over $\mathcal{X}$. For any positive integer $h$, we denote the sequence $\{a_1, \cdots, a_h\}$ by $a_{1:h}$. We use $\mathbb{1}\{\cdot\}$ to denote the indicator function.

## 2. Preliminaries

In this section, we present the single-step bandit formulation and the multi-step Markov decision process (MDP) formulation for LLMs.

**Bandit Formulation of LLMs.** A simple way to understand LLMs is through the bandit formulation. In this context, the prompt and the response are represented as $x \in \mathcal{X}$ and $y \in \mathcal{Y}$, respectively. Here, $\mathcal{X}$ refers to the set of prompts, while $\mathcal{Y}$ represents the set of responses. The LLM corresponds to the policy $\pi$ in this bandit framework, where $\pi(y \mid x)$ indicates the probability of generating the response $y$ given the prompt $x$.

**MDP Formulation of LLMs.** Following the notations in Zhong et al. (2024), we consider an MDP $\mathcal{M} = (\mathcal{S}, \mathcal{A}, \mathcal{P}, r, \rho, H)$. In this framework, $\mathcal{S}$ and $\mathcal{A}$ represent the state and action spaces, respectively. The transition kernel is denoted by $\mathcal{P} : \mathcal{S} \times \mathcal{A} \mapsto \Delta(\mathcal{S})$, while $r$ indicates the reward function. The initial distribution is defined by $\rho \in \Delta(\mathcal{S})$, and $H$ specifies the horizon length. A policy $\pi : \mathcal{S} \mapsto \Delta(\mathcal{A})$ maps a state to a distribution over the action space. Initially, an initial state is sampled using $s_1 \sim \rho$. At the $h$-th step, the agent receives the state $s_h$ and chooses the action $a_h \sim \pi(\cdot \mid s_h)$. This interaction continues until a specified ending condition is met, which will occur within $H$ steps.

In the context of generating large language models (LLMs), let $s_1 \sim \rho$ represent the prompt $x \sim \rho$. At each step $h$, the state $s_h = (s_1, a_{1:h-1})$ consists of the prompt $x$ and all tokens generated up to that point. The LLM acts as a policy $\pi$ that maps $s_h$ to a distribution over the action $a_h \sim \pi(\cdot \mid s_h)$, where the action signifies a token (or a series of consecutive tokens). The transition process is deterministic; it simply concatenates $s_h = (s_1, a_{1:h-1})$ and $a_h$ to create a new state $s_{h+1} = (s_1, a_{1:h})$. The generation process concludes with a special end-of-sentence token $\texttt{EoS}$, which will be generated within $H$ steps. For simplicity, we consider the length-$H$ trajectories $\{(s_h, a_h)\}_{h=1}^H$, noting that this does not lose generality since we can pad the $\texttt{EoS}$ token to the text to reach length $H$. With this notation and recognizing the autoregressive nature of LLMs, for any realizable trajectory $\{(s_h, a_h)\}_{h=1}^H$, the generation probability is given by $\pi(a_{1:H} \mid s_1) = \prod_{h=1}^H \pi(a_h \mid s_1, a_{1:h-1})$.

**Regularized Value Functions.** For a policy $\pi$, its entropy-regularized value function is defined as

$$V^\pi(s; r) = \mathbb{E}_\pi \left[ \sum_{h=1}^H \Big( r(s_h, a_h) - \beta \cdot \log \pi(a_h \mid s_h) \Big) \Big| s_1 = s \right], \tag{2.1}$$

where $\beta > 0$ is a regularization parameter. The regularized Q-function $Q^\pi$ of a policy $\pi$ is related to the regularized value function $V^\pi$ as

$$Q^\pi(s, a; r) = r(s, a) + \mathbb{E}_{s' \sim \mathcal{P}(\cdot \mid s, a)}[V^\pi(s'; r)],$$
$$V^\pi(s; r) = \mathbb{E}_{a \sim \pi(\cdot \mid s)}[-\beta \log \pi(a \mid s) + Q^\pi(s, a; r)], \tag{2.2}$$

The regularized optimal policy $\pi^*$ is the policy that maximizes the regularized value function defined in (2.1), and its corresponding optimal Q-function and value function are denoted as $Q^*$ and $V^*$, respectively. By (2.2), it can be shown that

$$V^*(s; r) = \log \sum_{a \in \mathcal{A}} \exp\big(Q^*(s, a; r)\big), \tag{2.3}$$

$$\pi^*(a \mid s) = \exp\{(Q^*(s, a; r) - V^*(s; r))/\beta\}$$
$$\propto \exp\big(Q^*(s, a; r)\big). \tag{2.4}$$

## 3. Unified Framework and Generic Algorithm

In this section, we present a new framework for LLM reasoning and our generic algorithm within this framework.

### 3.1. LLM as A Probabilistic Graphical Model

We consider four different spaces: $\mathcal{X}$ represents the prompt space, $\mathcal{Z}$ denotes the latent space that captures the intrinsic thought process (CoT), $\mathcal{Y}$ signifies the response space, and $\mathcal{O}$ stands for the evaluation signal space, which reflects the optimality of the prompt-latent-response tuple. Furthermore, for any $(x, z, y, o) \in \mathcal{X} \times \mathcal{Z} \times \mathcal{Y} \times \mathcal{O}$, we describe the generation process using the probabilistic graphical model illustrated in Figure 1. This indicates that

$$
\begin{aligned}
\mathbb{P}(z, y, o \,|\, x, \theta) &= \mathbb{P}(z, y \,|\, x, \theta) \cdot \mathbb{P}(o \,|\, x, z, y) \quad (3.1) \\
&= \mathbb{P}(z \,|\, x, \theta) \cdot \mathbb{P}(y \,|\, x, z, \theta) \cdot \mathbb{P}(o \,|\, x, z, y),
\end{aligned}
$$

where $\theta$ is the parameter of the LLM that guides the generation process. First, a latent variable $z$ is generated from the distribution $\mathbb{P}(\cdot \,|\, x, \theta)$, and then the response $y \sim \mathbb{P}(\cdot \,|\, x, z, \theta)$ is produced based on both the prompt $x$ and the latent variable $z$. Importantly, the probability $\mathbb{P}(o \,|\, x, z, y)$ is independent of the LLM parameterized by $\theta$, as we assume there exists a ground-truth judgment for the triplet $(x, z, y)$, such as a ground-truth/human reward function. Unlike traditional LLM frameworks that only consider the prompt space $\mathcal{X}$ and output space $\mathcal{Y}$, our framework incorporates both a latent thinking process space and an observation space. These additional components are crucial for mathematically understanding how to improve the quality of thinking processes using evaluation signals.

Under this probabilistic graphical modeling of LLMs, our learning objective is to maximize

$$
\mathcal{L}(\theta) = \log \mathbb{P}(z \in \mathscr{Z}, y \in \mathscr{Y}, o \in \mathscr{O} \,|\, x, \theta), \quad (3.2)
$$

where $\mathscr{Z} \subseteq \mathcal{Z}$, $\mathscr{Y} \subseteq \mathcal{Y}$, and $\mathscr{O} \subseteq \mathcal{O}$ denote the subsets of spaces representing the latent thinking process, response, and evaluation signals, respectively. In Section 3.2, we develop a general optimization algorithm that works with any choice of these spaces $(\mathscr{Z}, \mathscr{Y}, \mathscr{O})$. Subsequently, in Section 3.3, we show how this framework unifies existing learning approaches by demonstrating how different choices of these spaces $(\mathscr{Z}, \mathscr{Y}, \mathscr{O})$ correspond to various established learning paradigms and algorithms.

### 3.2. Bootstrapping Reinforced Thinking Process

We propose the algorithm, Bootstrapping Reinforced Thinking Process (BRiTE), to maximize the objective (3.2) within the framework proposed in the previous subsection. Since this objective may be difficult to optimize directly, we

rewrite it as

$$
\begin{aligned}
\mathcal{L}(\theta) &= \log \mathbb{P}(z \in \mathscr{Z}, y \in \mathscr{Y}, o \in \mathscr{O} \,|\, x, \theta) \\
&= \log \sum_{(z, y, o) \in \mathscr{Z} \times \mathscr{Y} \times \mathscr{O}} \mathbb{P}(z, y, o \,|\, x, \theta) \\
&= \max_{\mathbb{Q}(\cdot, \cdot, \cdot \,|\, x, \psi) \in \Delta(\mathscr{Z} \times \mathscr{Y} \times \mathscr{O})} \mathcal{L}_\psi(\theta), \quad (3.3)
\end{aligned}
$$

where $\mathbb{Q}(\cdot, \cdot, \cdot \,|\, x, \psi)$ can be regarded as another LM parametrized by $\psi$ and $\mathcal{L}_\psi(\theta)$ is

$$
\begin{aligned}
\mathcal{L}_\psi(\theta) = &\sum_{\substack{(z, y, o) \\ \in \mathscr{Z} \times \mathscr{Y} \times \mathscr{O}}} \log \mathbb{P}(z, y, o \,|\, x, \theta) \cdot \mathbb{Q}(z, y, o \,|\, x, \psi) \\
&- \sum_{\substack{(z, y, o) \\ \in \mathscr{Z} \times \mathscr{Y} \times \mathscr{O}}} \log \mathbb{Q}(z, y, o \,|\, x, \psi) \cdot \mathbb{Q}(z, y, o \,|\, x, \psi).
\end{aligned}
$$

The last equality of (3.3) follows the following lemma:

**Lemma 3.1.** *For any set $\mathcal{W}$ and non-negative numbers $\{P_w \geq 0\}_{w \in \mathcal{W}}$, it holds*

$$
\log \sum_{w \in \mathcal{W}} P_w = \max_{q \in \Delta(\mathcal{W})} \mathbb{E}_{w \sim q(\cdot)}[\log P_w - \log q(w)].
$$

*The maximum is achieved when $q(w) = P_w / (\sum_{w' \in \mathcal{W}} P_{w'})$.*

*Proof of Lemma 3.1.* This lemma is equivalent to the non-negativity of the KL divergence. For a detailed proof, please refer to Appendix C.1. □

We have transformed the problem of maximizing $\mathcal{L}(\theta)$ into maximizing its lower bound $\mathcal{L}_\psi$. This shares the same spirit with the Expectation-Maximization (EM) algorithm (Dempster et al., 1977) and Variational Autoencoders (VAE) (Kingma, 2013). To solve this optimization problem, we iteratively update the parameters $\psi$ and $\theta$ to maximize (3.3). Given $\psi_t$ and $\theta_t$, the updating rules of $\psi_{t+1}$ and $\theta_{t+1}$ are specified as

- choosing $\psi_{t+1}$ such that

$$
\begin{aligned}
\mathbb{Q}(z, y, o \,|\, x, \psi_{t+1}) &= \operatorname*{argmax}_{\mathbb{Q}(\cdot, \cdot, \cdot \,|\, x, \psi)} \mathcal{L}_\psi(\theta_t) \quad (3.4) \\
&= \frac{\mathbb{P}(z, y, o \,|\, x, \theta_t)}{\sum_{(z, y, o) \in \mathscr{Z} \times \mathscr{Y} \times \mathscr{O}} \mathbb{P}(z, y, o \,|\, x, \theta_t)} \propto \mathbb{P}(z, y, o \,|\, x, \theta_t).
\end{aligned}
$$

This is implied by the optimality condition in Lemma 3.1.

- choosing $\theta_{t+1}$ such that

$$
\begin{aligned}
\theta_{t+1} &= \operatorname*{argmax}_\theta \mathcal{L}_{\psi_{t+1}}(\theta) \quad (3.5) \\
&= \operatorname*{argmax}_\theta \left\{ \sum_{\substack{(z, y, o) \\ \in \mathscr{Z} \times \mathscr{Y} \times \mathscr{O}}} \log \mathbb{P}(z, y, o \,|\, x, \theta) \cdot \mathbb{Q}(z, y, o \,|\, x, \psi_{t+1}) \right\} \\
&= \operatorname*{argmax}_\theta \left\{ \sum_{\substack{(z, y, o) \\ \in \mathscr{Z} \times \mathscr{Y} \times \mathscr{O}}} \log \mathbb{P}(z, y \,|\, x, \theta) \cdot \mathbb{Q}(z, y, o \,|\, x, \psi_{t+1}) \right\},
\end{aligned}
$$

where the second equality is implied by (3.1).

Intuitively, during the $\psi$-updating step in (3.4), the proposed algorithm learns to generate high-quality thinking processes by focusing on verified correct responses. Specifically, when we define $\mathcal{O} = \{O^*\}$, where $O^*$ represents the correct evaluation signal in mathematical tasks, $\mathbb{P}(z, y, O^* \mid x)$ represents the probability distribution of generating both a high-quality thinking process $z$ and correct final answer $y$. Following this, in the following $\theta$-updating step in (3.5), the algorithm fine-tunes the Large Language Model by maximizing the joint distribution between the learned thinking process generator and the LLM's output with respect to parameter $\theta$. This fine-tuning approach encompasses several learning paradigms and algorithms, including SFT, RLHF, and rejection sampling-based algorithms, as detailed in Examples 3.4, 3.5, and 3.6. The two-stage BRiTE algorithm draws inspiration from the classical EM algorithm. While the probabilistic graphical model used in BRiTE differs from traditional latent variable models, the $\psi$-updating and $\theta$-updating steps correspond to the E-step and M-step in the EM algorithm, respectively.

Before presenting the theoretical results for BRiTE, we make the following assumption about the generation probability of parametrized LLMs.

**Assumption 3.2.** Assume that LLM is parameterized by $\theta$, we denote that the logits of generating $(z, y)$ conditioned on $x$ by $f_\theta(x, z, y)$, then the probability of $\mathbb{P}(z, y \mid x, \theta)$ takes the form

$$\mathbb{P}(z, y \mid x, \theta) = \exp\left(f_\theta(x, z, y) - A(x, \theta)\right)$$
$$\propto \exp\left(f_\theta(x, z, y)\right),$$

where $A_\theta(x, \theta)$ is the normalization factor. Moreover, we assume that $f_\theta \in \mathcal{H}$ for some reproducing kernel Hilbert space (RKHS)[1] associated with the kernel $\mathcal{K} : (\mathcal{X} \times \mathcal{Z} \times \mathcal{Y}) \times (\mathcal{X} \times \mathcal{Z} \times \mathcal{Y}) \mapsto \mathbb{R}$.

The $f_\theta(x, z, y)$ in Assumption 3.2 represents the logits for predicting $(y, z)$ based on the prompt $x$, thereby capturing the current generation method of modern transformer-based LLMs. With this assumption, we establish the convergence result of our algorithm in the following theorem, whose proof is deferred to Appendix D.1.

**Theorem 3.3.** Suppose Assumption 3.2 holds. Given that $\mathcal{L}$ is concave and $\theta^* = \operatorname{argmax}_\theta \mathcal{L}(\theta)$, we have

$$\min_{1 \le t \le T} \{\mathcal{L}(\theta^*) - \mathcal{L}(\theta_t)\} \le \frac{\mathrm{KL}\left(\mathbb{P}(\cdot, \cdot \mid x, \theta_1) \| \mathbb{P}(\cdot, \cdot \mid x, \theta^*)\right)}{T},$$

[1] We say $\mathcal{H}$ is a RKHS on the set $\mathcal{W}$ with the reproducing kernel $\mathcal{K} : \mathcal{W} \times \mathcal{W} \mapsto \mathbb{R}$ if the inner product $\langle \cdot, \cdot \rangle$ satisfies $f(w) = \langle \mathcal{K}(w, \cdot), f \rangle$, for any $(f, w) \in \mathcal{H} \times \mathcal{W}$.

Theorem 3.3 establishes a convergence rate of $1/T$, where $T$ represents the number of iteration steps. In our analysis, we link our algorithm to classical mirror descent algorithms (Nemirovskij & Yudin, 1983; Bubeck et al., 2015), which have also been utilized in recent studies of policy optimization algorithms (Agarwal et al., 2021; Cai et al., 2020; Zhong & Zhang, 2024). Additionally, our results rely on a concave assumption. This assumption is crucial for proving global convergence results, and we can still obtain a weaker guarantee for stationary points even without this concave condition; see Appendix D.2 for details.

Having established a theoretical guarantee for our proposed generic algorithm, we will demonstrate how our algorithm can incorporate many existing learning algorithms, highlighting the broad applicability of our algorithmic framework and its theoretical guarantee.

### 3.3. Connections to Existing Learning Paradigms and Algorithms

**Example 3.4** (Pre-training, SFT and Conditional SFT). *We first set aside the latent space and observation space, meaning that $\mathcal{Z} = \mathcal{O} = \emptyset$. In this case, our learning objective (3.2) encompasses two key processes: (I) pre-training, where we let $x$ represent the prompt and $\mathcal{Y}$ the next token; and (II) supervised fine-tuning (SFT), where we define $\mathcal{Y} = \{y^*(x)\}$ as the expert response corresponding to the prompt $x$. Moreover, for the prompt-response pair $(x, y)$ and singleton space $\mathcal{Y} = \{y\}$, we consider $\mathcal{Z} = \mathbb{R}^+$ and $\mathcal{Z} = \{R(x, y)\}$ be the reward corresponding to the prompt-response pair, then our objective recovers the conditional SFT (Lu et al., 2022; Dong et al., 2023b; Yang et al., 2024).*

**Example 3.5** (RLHF: PPO and DPO). *We choose $\mathcal{Y} = \mathcal{Y}$ and $\mathcal{Z} = \mathcal{Z}$ as the complete response space and latent space, respectively. Additionally, we set $\mathcal{O} = \{0, 1\}$, where $1$ indicates optimality and $0$ indicates non-optimality, respectively. Since our goal is to maximize the probability of observing the signal of optimality, we focus on $\mathcal{O} = \{1\}$. We also assume that $\mathbb{P}(o = 1 \mid x, z, y) = \exp(R(x, z, y)/\beta)$ for some reward function $R$ and $\beta > 0$. With these choices, we have*

$$\mathcal{L}_\psi(\theta) = \sum_{(z,y) \in \mathcal{Z} \times \mathcal{Y}} \log \mathbb{P}(z, y, 1 \mid x, \theta) \cdot \mathbb{Q}(z, y, 1 \mid x, \psi)$$
$$- \sum_{(z,y) \in \mathcal{Z} \times \mathcal{Y}} \log \mathbb{Q}(z, y, 1 \mid x, \psi) \cdot \mathbb{Q}(z, y, 1 \mid x, \psi)$$
$$= \mathbb{E}_{(z,y) \sim \mathbb{Q}(\cdot, \cdot \mid x, \psi)} \left[ R(x, z, y) - \beta \log \frac{\mathbb{Q}(z, y \mid x, \psi)}{\mathbb{P}(z, y \mid x, \theta)} \right],$$

*where $\mathbb{Q}(z, y \mid x, \psi) = \sum_{o \in \mathcal{O}} \mathbb{Q}(z, y, o \mid x, \psi) = \mathbb{Q}(z, y, 1 \mid x, \psi)$. This expression recovers the proximal policy optimization (PPO; Schulman et al., 2017) for RLHF (Christiano et al., 2017; Ouyang et al., 2022), where $\psi$ represents the LLM being optimized, and $\theta$ stands for the reference policy. Assuming that the preference data $\{(x, z^+, y^+, z^-, y^-)\}$ is drawn from the Bradley-Terry (BT)*

model ([Bradley & Terry, 1952](#)), with $(z^+, y^+)$ denoting preferred data and $(z^-, y^-)$ indicating dispreferred data, one can follow [Rafailov et al. (2024)](#) to derive the latent direct preference optimization (latent DPO) objective:

$$\mathcal{L}_{\text{Latent}-\text{DPO}} \quad (3.6)$$

$$= \sigma \left( \beta \log \frac{\mathbb{Q}(z^+, y^+ \mid x, \psi)}{\mathbb{P}(z^+, y^+ \mid x, \theta)} - \beta \log \frac{\mathbb{Q}(z^-, y^- \mid x, \psi)}{\mathbb{P}(z^-, y^- \mid x, \theta)} \right),$$

where $\sigma$ is the sigmoid function. In contrast to the standard DPO objective in [Rafailov et al. (2024)](#), the objective in (3.6) additionally incorporates the latent variable $z$.

**Example 3.6** (Rejection Sampling EM Methods). *Let $\mathscr{Y} = \mathcal{Y}$ represent the complete response space, and define $\mathscr{O} = \{0, 1\}$, where 1 indicates the optimal outcome and 0 indicates otherwise. We focus on the optimal signal, denoted as $\mathscr{O} = \{1\}$. Additionally, we denote $\mathscr{Z} = \mathcal{Z}$ as the complete thinking process space. The updating rule in (3.4) is given by $\mathbb{Q}(z, y, 1 \mid x, \psi_{t+1}) \propto \mathbb{P}(z, y, 1 \mid x, \theta_t) = \mathbb{P}(z, y \mid x, \theta_t) \cdot \mathbb{P}(o = 1 \mid x, z, y)$. Consequently, the update in (3.5) can be expressed as*

$$\theta_{t+1} = \arg\max_{\theta} \left\{ \sum_{(z,y) \in \mathcal{Z} \times \mathcal{Y}} \log \mathbb{P}(z, y \mid x, \theta) \cdot \mathbb{Q}(z, y, 1 \mid x, \theta_t) \right\}$$

$$= \arg\max_{\theta} \left\{ \mathbb{E}_{(z,y) \sim \mathbb{P}(\cdot, \cdot \mid x, \theta_t)} \left[ \log \mathbb{P}(z, y \mid x, \theta) \cdot \mathbb{P}(1 \mid x, z, y) \right] \right\}$$
$$(3.7)$$

1. *If we simplify the scenario by omitting the latent space (i.e., $\mathcal{Z} = \emptyset$) and assume that $\mathbb{P}(o = 1 \mid x, z, y) = \mathbb{1}\{y \text{ is the correct answer}\}$, this leads to the STaR algorithm ([Zelikman et al., 2022](#)) or rejection sampling fine-tuning ([Dong et al., 2023a](#); [Yuan et al., 2023](#); [2024](#)), which performs supervised fine-tuning after rejection sampling based on verifier results.*

2. *If we simplify the scenario by omitting the latent space (i.e., $\mathcal{Z} = \emptyset$) and assuming $\mathbb{P}(o = 1 \mid x, y) = \exp(R(x, y)/\beta)$ for some true reward function $R$, the updating rule in (3.7) simplifies to*

$$\theta_{t+1} = \arg\max_{\theta} \left\{ \mathbb{E}_{y \sim \mathbb{P}(\cdot \mid x, \theta_t)} \left[ \log \mathbb{P}(y \mid x, \theta) \right. \right.$$
$$\left. \left. \cdot \exp \left( R(x, y)/\beta \right) \right] \right\}. \quad (3.8)$$

*This recovers the ReST$^{\text{EM}}$ algorithm presented in ([Singh et al., 2023](#)).*

Finally, we note that [Neal & Hinton (1998)](#) first introduced the unified view of rejection sampling EM, which [Rush & Ritter (2024)](#) later expanded upon.

By combining Theorem 3.3 with the examples in this section, we provide theoretical guarantees for these learning algorithms. As a result, we establish theoretical foundations for two approaches: (1) PPO, connecting to the work of ([Schulman et al., 2017](#); [Cai et al., 2020](#); [Zhong & Zhang, 2024](#)), and (2) Generalized Rest$^{\text{EM}}$ in Example 3.6 (which includes vanilla Rest$^{\text{EM}}$ ([Singh et al., 2023](#)) as a special case), presenting the first theoretical analysis of its kind.

## 3.4. Practical Implementation: The Power of Reinforcement Learning

We have demonstrated that our generic algorithm encompasses a wide range of existing learning paradigms and algorithms, with a provable convergence guarantee. In this section, we carefully examine the practicality of our algorithm.

First, the $\theta$-updating step is relatively straightforward to implement, as this step simply maximizes the predicted probability of the next tokens after we sample $(z, y, o)$ from $\mathbb{Q}(z, y, o \mid x, \psi_{t+1})$. In contrast, the $\psi$-updating step can be challenging in certain contexts. For instance, when $\mathscr{Y} = \{y\}$ represents the response corresponding to $x$, $\mathscr{O} = \{1\}$ indicates the optimality signal of interest, and $\mathscr{Z} = \mathcal{Z}$, we have $\mathbb{Q}(z, y, 1 \mid x, \theta) \propto \mathbb{P}(z, y, 1 \mid x, \theta) = \mathbb{P}(z \mid x, y, 1, \theta)$—the posterior of the latent variable $z$. Intuitively, obtaining this distribution requires us to identify the ideal latent (CoT) based on the pair $(x, y)$, which represents an intractable posterior.

To achieve this goal, we aim to use RL to train an LLM that characterizes the distribution $\mathbb{Q}(z, y, o \mid x, \psi_{t+1})$ or $\mathbb{P}(z, y, o \mid x, \theta_t)$ in (3.4). Since an LLM acts as the policy of an MDP, our approach involves two main steps: (i) constructing an MDP whose optimal policy matches the distribution $\mathbb{Q}$ that we need to learn; and (ii) applying RL algorithms to solve this MDP and identify the optimal policy to learn the desired distribution. These two steps convert the challenging sampling problem to a more amenable RL optimization problem. Notably, the main challenge we face is *reward shaping*, which involves designing appropriate reward functions to ensure that the optimal policy aligns with the intended LLM that accurately represents the posterior. Our approach to reward shaping is based on the following proposition, which characterizes the optimal policy for deterministic entropy-regularized MDPs.

**Proposition 3.7.** *Assuming the transition dynamic of entropy-regularized MDP is deterministic, then for any trajectories $\{(s_i, a_i)\}_{i=h}^{H}$ satisfying $a_H = \texttt{EOS}$, we have*

$$\pi^*\left(a_h \cup \{(s_i, a_i)\}_{i=h+1}^{H} \mid s_h\right) \propto \exp\left(\frac{1}{\beta} \sum_{i=h}^{H} r(s_i, a_i)\right).$$

The proof is deferred to Appendix C.2. By Proposition 3.7, if we select $\beta = 1$ and use the total reward as $\log \mathbb{P}(z, y, o \mid x, \theta_t)$, the resulting optimal policy recovers $\mathbb{Q}$ defined in (3.4). This choice of total reward function can naturally be assigned to each token as the token-reward function, due to the autoregressive nature of LLM generation. Specifically, when $(z, y, o)$ is represented by a token sequence $a_{1:\tau}$, the total reward $\log \mathbb{P}(z, y, o \mid x, \theta_t)$ can be expressed as $\sum_{j=1}^{\tau} \log \mathbb{P}(a_j \mid a_{1:j-1}, x)$, where $\log \mathbb{P}(a_j \mid a_{1:j-1}, x)$ serves as the reward for the $j$-th token in RL training.

Finally, we remark that the application of RL to text generation has been explored in previous studies, such as Guo et al. (2021). However, there appears to be no work applying RL to generate the thinking process to enhance reasoning capabilities within the context of LLMs. Additionally, existing studies do not address the issue of reward shaping, which is a crucial problem in our scenario and is tackled by Proposition 3.7.

## 4. Experiments

In this section, we systematically demonstrate that our unified algorithm enhances the reasoning capability of LLMs.

### 4.1. Experimental Setups

**Tasks and Datasets.** To evaluate mathematical reasoning capabilities, we conduct experiments on two prominent benchmarks: GSM8K (Cobbe et al., 2021) and MATH (Hendrycks et al., 2021). GSM8K contains 1,319 high-quality grade school math problems requiring multi-step reasoning, ranging from basic arithmetic to elementary algebra. The MATH dataset comprises 5,000 competition-level problems covering advanced topics such as algebra, geometry, number theory, probability, and calculus. These problems require more sophisticated problem-solving strategies and formal mathematical reasoning. The training sets of GSM8K and MATH datasets each contain approximately 7,500 data points. Each data point includes a question $x$, along with human-annotated rationale $z^*$ and the correct final answer $y^*$.

**Base Models.** We use several open-source instruction-tuned LLMs as base models, including Gemma-2-9b-it (Team et al., 2024b), Gemma-1.1-7B-it (Team et al., 2024a), Mixtral-7B-Instruct-v0.2 (Jiang et al., 2023) and Llama-3-8B-Instruct (Touvron et al., 2023).

**Baselines.** Our first baseline is rejection sampling (**RS**) methods (Neal & Hinton, 1998; Dong et al., 2023a; Yuan et al., 2023; Zelikman et al., 2022). For each problem $x$, we generate $N = 30$ candidate rationales and select one ($z_{rs}$) containing the correct answer $y^*$. The model is fine-tuned on these problem-rationale-answer tuples $\{(x, z_{rs}, y^*)\}$. Importantly, this method does not use human-annotated rationales $z^*$, making it directly comparable to our approach. We also test the performance of supervised fine-tuning (**SFT**) on datasets with human-annotated rationales $\{(x, z^*, y^*)\}$. As a baseline learning from preference data, we implement the **iterative DPO** (Xiong et al., 2024a; Pang et al., 2024), initialized with instruction-tuned models. The training process consists of 3 iterations. In each iteration, we: (1) use the current model to generate 30 CoT and responses $\{(z, y)\}$ per prompt; (2) select the best and worst response pairs

$(z^+, y^+, z^-, y^-)$ based on the correctness of the $y$; and (3) apply DPO training (see Rafailov et al. (2024) or (3.6)) on these preference tuples $\{(x, z^+, y^+, z^-, y^-)\}$.

**Implementations of BRiTE.** To systematically demonstrate the effectiveness of our algorithm, we implemented BRiTE in three distinct configurations:

1. We implement the $\psi$-updating step of BRiTE to obtain $\mathbb{Q}$ using (3.4), examining how the generated thinking process aligns with the desired reasoning path to enhance LLM reasoning capabilities. Following our theoretical framework in Section 3.4, we optimize two types of reward functions in the entropy-regularized MDP: (i) $\log(z, y, o \mid x) = \log \mathbb{P}(z, y^*(x) \mid x)$, where $y^*$ represents the correct answer for question $x$. This equality holds because we focus solely on correct questions, using proper choices of $\mathcal{O} = \{\text{answer verified to be correct}\}$ and $\mathcal{Y} = \{y^*\}$. (ii) $\log(z, y, o \mid x) = \log \mathbb{P}(z, y \mid x) + R(x, z, y)/\beta$, where we assume that $\mathbb{P}(o = 1 \mid x, z, y) = \exp(R(x, z, y)/\beta)$ with $\mathcal{O} = \{1\}$. We employ PPO (Schulman et al., 2017) or GRPO (Shao et al., 2024) to optimize this MDP.

2. BRiTE first obtains $\mathbb{Q}$ in (3.4) through RL, then updates $\theta$ in (3.5) using SFT. The SFT data consists of three components: problems $x$, thinking processes $z_{\mathbb{Q}}(x)$ generated by $\mathbb{Q}$, along with the ground truth answers $y^*$.

3. We implement BRiTE-DPO, which consists of 3 iterations. During the $t$-th iteration, we first learn the distribution $\mathbb{Q}$ according to (3.4). Then for each prompt $x$, we leverage $\mathbb{Q}$ to generate $N = 30$ reasoning process and output pairs $\{(z_{\mathbb{Q}}, y_{\mathbb{Q}})\}$. Subsequently, we evaluate the correctness of each $y_{\mathbb{Q}}$ to identify the best and worst latent reasoning processes and construct response pairs $(z_{\mathbb{Q}}^+, y_{\mathbb{Q}}^+, z_{\mathbb{Q}}^-, y_{\mathbb{Q}}^-)$ for use in DPO training (3.6).

### 4.2. Experimental Result and Analysis

**1. BRiTE Significantly Improves Existing Rejection Sampling Fine-Tuning Algorithms.** We begin by comparing BRiTE-SFT with rejection sampling EM-type algorithms, both of which aim to enhance the reasoning capabilities of LLMs by bootstrapping the thinking process. The key distinction is that BRiTE-SFT uses RL for this bootstrapping, while rejection sampling methods rely on sampling techniques. From Table 1, we observe that BRiTE-SFT consistently outperforms rejection sampling-based algorithms across all models, achieving a concrete accuracy improvement of 1-10 points. These improvements are primarily attributed to BRiTE-SFT's ability to generate higher-quality thinking processes compared to rejection sampling, highlighting the potential of RL-driven approaches to advance

| Algorithm | Mistral-7B-Instruct-v0.2 | | Gemma-1.1-7B-it | | Gemma-2-9B-it | | Llama-3-8B-Instruct | |
|---|---|---|---|---|---|---|---|---|
| | GSM8K | MATH | GSM8K | MATH | GSM8K | MATH | GSM8K | MATH |
| — | 41.8 | 9.8 | 49.0 | 18.8 | 81.3 | 37.3 | 79.2 | 28.3 |
| SFT | **52.8** | **13.6** | 57.5 | 19.6 | 80.1 | 41.5 | 72.6 | 27.1 |
| RS | 47.7 | 10.3 | 58.4 | 18.7 | 87.6 | 47.5 | 79.5 | 28.9 |
| BRiTE | 52.2 | 11.2 | **59.2** | **23.7** | **89.7** | **50.5** | **81.0** | **30.0** |

*Table 1.* A comparison of the performance of three algorithms: BRiTE, rejection sampling (RS) type algorithms, and SFT using human-annotated data.

LLM reasoning through more effective bootstrapping of the thinking process.

## 2. BRiTE Matches or Even Enhances the Performance of SFT with Human-Annotated Thinking Process. To
further validate the effectiveness of BRiTE's RL-based bootstrapping mechanism, we compare its performance against SFT using human-annotated thinking process data. Human annotations are widely regarded as the gold standard for training LLMs, as they include explicitly crafted reasoning pathways to ensure high-quality outputs. However, as shown in Table 1, BRiTE achieves performance on par with, and in some cases surpasses, that of human-annotated reasoning-based fine-tuning in downstream tasks. This highlights a remarkable outcome: RL-generated thinking processes can achieve a quality comparable to or even superior to human-derived reasoning. This somewhat surprising result underscores the value of BRiTE as a cost-effective alternative to labor-intensive and time-consuming human annotation processes. By reducing reliance on manual annotation, BRiTE sheds light on mitigating the bottleneck associated with creating high-quality datasets, particularly for complex reasoning tasks where human annotations can be prohibitively expensive or inconsistent.

## 3. BRiTE Further Enhances the Reasoning Capacity in RLHF Stage. In addition to advancing reasoning algorithms using question-(rational)-answer data, BRiTE demonstrates the potential to enhance RLHF algorithms that rely on preference data. As shown by the experimental results in Figure 2, BRiTE consistently outperforms iterative DPO across multiple benchmarks, highlighting its effectiveness in the RLHF stage. This superior performance is attributed to BRiTE's ability to facilitate more structured and contextually nuanced reasoning processes, which, in turn, produce higher-quality preference data and enable more robust policy refinement. These findings not only underscore the versatility of BRiTE but also affirm its value in optimizing RLHF-based fine-tuning pipelines, further cementing its role as a generic algorithm for enhancing LLM reasoning in the post-training stage.

## 4. BRiTE Can Also Improve Code Generation Abil-

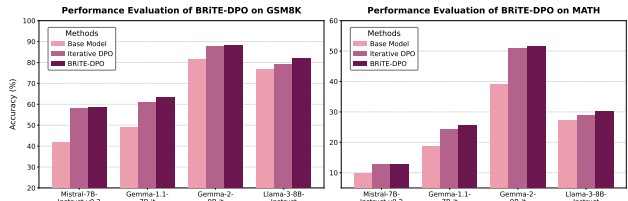

*Figure 2.* Comparison between BRiTE and iterative DPO.

| Algorithm | HumanEval | | BCB (Instruct) | |
|---|---|---|---|---|
| | Basic (%) | Plus (%) | Hard (%) | Full (%) |
| — | 78.0 | 70.7 | 10.1 | 35.5 |
| SFT | 78.0 | 67.7 | 11.5 | **37.2** |
| RS | 79.3 | **73.2** | 11.5 | 35.6 |
| BRiTE | **81.7** | 72.6 | **15.5** | 36.3 |

*Table 2.* Results of BRiTE on coding generation task using the deepseek-coder-6.7b-instruct model.

**ity.** Finally, we extend the evaluation of BRiTE beyond mathematics tasks to assess its performance on code generation. The results, presented in Table 2, demonstrate a consistent trend observed in mathematics tasks: BRiTE outperforms rejection sampling-based algorithms and even surpasses SFT using human-annotated answers. This highlights BRiTE's versatility and effectiveness across diverse problem domains, showcasing its ability to generalize to a wide range of reasoning tasks. We also remark that rejection sampling-based algorithms require the code datasets to be equipped with correct unit tests, which are unnecessary for BRiTE. For a detailed description of the experimental setup and configuration, refer to Appendix E.2.

### 4.3. Enhanced Math Reasoning via Expanded Dataset and Advanced Base Model

In the previous subsection, BRiTE's performance did not demonstrate significant advantages over other algorithms like rejection sampling. This was primarily due to two factors: the instruct models had already undergone post-training, and the dataset size was limited. To overcome

| Method | MATH500 | Minerva Math | OlympiadBench | AIME24 | AMC23 | GPQA Diamond |
|---|---|---|---|---|---|---|
| — | 44.1 | 12.9 | 16.1 | 0.9 | 10.1 | 25.9 |
| RS | 54.3 | 21.0 | 23.1 | 5.6 | 31.6 | 26.9 |
| BRiTE ($\psi$-update) | 79.1 | 35.0 | 35.7 | 14.3 | 57.7 | 28.5 |
| BRiTE ($\theta$-update) | 76.9 | 40.6 | 37.0 | 14.4 | 57.1 | 29.8 |
| BRiTE-iter-2 ($\psi$-update) | **80.6** | **41.3** | 37.3 | 14.3 | **57.9** | 29.9 |
| BRiTE-iter-2 ($\theta$-update) | 78.2 | 39.8 | **37.9** | **15.3** | 56.4 | **30.1** |

*Table 3.* Performance comparison across different reasoning benchmarks.

these constraints, we have implemented a larger training dataset based on the more advanced Qwen base model.

Specifically, we perform BRiTE on the model Qwen2.5-7B (Team, 2024) with a mixed dataset (the size is around 40K) of `RUC-AIBOX/STILL-3-Preview-RL-Data`[2], MATH, and historical AIME problems (excluding AIME 2024). We evaluated models trained using the RS baseline, the $\psi$-update and $\theta$-update of BRiTE on a diverse set of reasoning benchmarks, including six challenging math and science reasoning benchmarks: Math-500, Minerva Math, Olympiad Bench, AIME24, AMC23, and GPQA Diamond. Here MATH500 contains 500 competition-level math problems drawn from the MATH dataset (Hendrycks et al., 2021), spanning subjects from algebra to precalculus and accompanied by solutions . Minerva Math comprises 272 undergraduate-level quantitative problems in physics, chemistry, biology, economics, and mathematics, designed to test advanced multi-step reasoning (Lewkowycz et al., 2022). OlympiadBench (He et al., 2024) is a collection of 8,952 Olympiad-caliber problems in math and physics (with 57% including diagrams) provided in both English and Chinese, each with a detailed solution . Drawing from its extensive entries, we selected 674 open-ended, competition-style, text-only problems from Olympiad-Bench. AMC23 (40 problems) and AIME24 (30 problems) are benchmarks based on recent AMC 12 (2023) and AIME (2024) contests (Mathematical Association of America, 2023; 2024), representing high school-level competition questions of moderate and high difficulty, respectively . Finally, GPQA Diamond (Rein et al., 2024) consists of 198 graduate-level scientific questions verified by experts. Each of these datasets challenges models with a different profile of difficulty and domain coverage, collectively evaluating a broad spectrum of mathematical reasoning capabilities. We evaluate the pass@1 accuracy (64 sampling times for AIME24 and AMC23 and 8 sampling times for the others) for each benchmark.

As shown in Table 3, BRiTE with an external verifier can improve reject sampling (RS) significantly. Specifically, on MATH500, Minerva Math and AMC23, BRiTE improve upon RS by over 15 points in accuracy, showcasing the

---

[2] https://huggingface.co/datasets/RUC-AIBOX/STILL-3-Preview-RL-Data

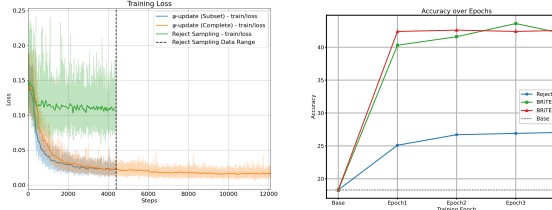

*Figure 3. Left:* Training dynamics of BRiTE with an external verifier and rejection sampling. *Right:* Mean accuracy of benchmark scores of models trained by BRiTE with an external verifier and reject sampling during the training process.

strong impact of RL-bootstrapped rationales. Moreover, we also show that the iterative training version of BRiTE leads to some improvements, though the training in iteration 1 reaches a plateau that may limit further gains. Figure 3 (left) shows that BRiTE exhibits faster and more stable training loss reduction compared to rejection sampling, indicating more effective use of feedback during optimization. Figure 3 (right) further confirms this trend across benchmarks: BRiTE consistently achieves higher mean accuracy throughout training. These results indicate that BRiTE not only improves final performance, but also accelerates the acquisition of reasoning capabilities during training.

## 5. Conclusion

In this work, we explore methods for enhancing language model reasoning through the automated generation of high-quality thinking processes. We present a unified probabilistic framework that characterizes both the reasoning process and evaluation signals. Within this framework, we develop the Bootstrapping Reinforced Thinking Process (BRiTE) algorithm, which advances automated reasoning generation through reinforcement learning during inference and incorporates improved reasoning processes into post-training phases. Furthermore, we demonstrate that BRiTE possesses a provable convergence property and unifies various existing learning paradigms and algorithms. Extensive empirical results on mathematics tasks show that our approach surpasses traditional chain-of-thought prompting while enhancing existing supervised/rejection sampling fine-tuning and reinforcement learning from human feedback methods.

## Impact Statement

This paper presents work whose goal is to advance the field of Machine Learning. There are many potential societal consequences of our work, none which we feel must be specifically highlighted here.

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

## A. Author Contributions

This work stems from the valuable contributions and close collaboration of all authors. The first seven authors and ZW are the core contributors to this project, all participating in discussions of problem formulation and theory, as well as experimental implementation. In particular, HZ and ZW lead the project, primarily propose the methodology, derive the theoretical results, guide experimental progress, contribute to early baselines, and write the paper. YY, SZ, and XX mainly contribute to the initial implementation of BRiTE without verifier reward for Llama, Gemma, and Mistral models, while YL, YZ, and ZL mainly contribute to BRiTE with verifier reward for larger datasets and Qwen base models. In addition, YL, YZ, and ZL make key contributions to direct preference learning, rejection sampling, and code generation, respectively. Other authors also make significant contributions to this work, providing computational resources and offering suggestions for theoretical analysis, experiment design, and paper writing.

## B. Additional Related Works

**RLHF.** RLHF (Christiano et al., 2017; Ziegler et al., 2019), also known as the dueling RL (Yue et al., 2012; Pacchiano et al., 2021) or preference-based RL (Wirth et al., 2017; Chen et al., 2022), has emerged as a breakthrough technique in language model development, playing a pivotal role in the success of ChatGPT (Ouyang et al., 2022) and subsequent large language models by effectively aligning model outputs with human preferences. In the standard RLHF pipeline, the first stage involves learning a reward function from human preference data, followed by optimization using proximal policy optimization (Schulman et al., 2017). However, this approach demands substantial computational resources and requires careful tuning of numerous hyperparameters. Direct preference learning (e.g., Zhao et al., 2023; Rafailov et al., 2024; Azar et al., 2024; Tang et al., 2024; Meng et al., 2024; Liu et al., 2024b; Cen et al., 2024) offers an alternative by learning the desired policy directly, circumventing the need for explicit reward learning and optimization. Recent work by Xiong et al. (2024a); Xie et al. (2024); Cen et al. (2024); Zhang et al. (2024) extends these offline approaches through online iterative learning to gather on-policy data for enhanced performance. However, these algorithms mainly focus on the chat task and do not explicitly model the latent reasoning process, thus not fully extracting the reasoning power of LLMs. While Pang et al. (2024); Wu et al. (2024) incorporate reasoning processes into DPO training using CoT prompting, our approach distinctively generates reasoning processes automatically through RL, demonstrating significant improvements in model reasoning capacity. Some work (e.g., Xiong et al., 2024b) combines iterative DPO with tool use to improve reasoning capabilities, which is beyond the scope of our current work.

## C. Missing Proofs in the Main Paper

### C.1. Proof of Lemma 3.1

*Proof of Lemma 3.1.* For any $q(\cdot) \in \Delta(\mathcal{W})$, we have

$$\mathbb{E}_{w \sim q(\cdot)}[\log P_w - \log q(w)] - \log \sum_{w \in \mathcal{W}} P_z = \mathbb{E}_{w \sim q(\cdot)}\left[\log \frac{P_w}{\sum_{w' \in \mathcal{W}} P_{w'}} - \log q(w)\right] = -\mathrm{KL}(q\|\widetilde{p}) \leq 0,$$

where $\widetilde{p}$ is the distribution defined as $\widetilde{p}(w) = P_w/(\sum_{w' \in \mathcal{W}} P_{w'})$, and the equality is achieved when $q = \widetilde{p}$. Hence, we have finished the proof of Lemma 3.1. □

### C.2. Proof of Proposition 3.7

*Proof of Proposition 3.7.* The first equation follows from the deterministic transition. We will now focus on proving the second propositional relationship. According to equation (2.3), we have:

$$\pi^*(a_i \,|\, s_i) = \exp\{(Q^*(s_i, a_i) - V^*(s_i))/\beta\}, \qquad \forall h \leq i \leq H,$$

which implies that

$$
\begin{aligned}
\sum_{i=h}^{H} \beta \log \pi^*(a_i \,|\, s_i) &= \sum_{i=h}^{H} \big(Q^*(s_i, a_i) - V^*(s_i)\big) \\
&= \sum_{i=h}^{H-1} \big(r(s_i, a_i) + V^*(s_{i+1}) - V^*(s_i)\big) + \big(r(s_H, a_H) - V^*(s_H)\big) \\
&= \sum_{i=h}^{H} r(s_i, a_i) - V^*(s_h),
\end{aligned}
\tag{C.1}
$$

where the second equality uses the fact that $a_H = \texttt{EOS}$. Hence, we have

$$
\prod_{i=h}^{H} \pi^*(a_i \,|\, s_i) = \frac{\exp\big(\frac{1}{\beta} \sum_{i=h}^{H} r(s_i, a_i)\big)}{\exp(V^*(s_h)/\beta)} \propto \exp\Big(\frac{1}{\beta} \sum_{i=h}^{H} r(s_i, a_i)\Big),
$$

where the last step is obtained by the fact that $s_h$ is a fixed state, independent of $a_h \cup \{(s_i, a_i)\}_{i=h+1}^{H}$.  □

## D. Convergence Results

### D.1. Proof of Theorem 3.3

Before starting the proof of Theorem 3.3, we present two technical lemmas.

**Lemma D.1.** *For any $(\theta, \theta')$ and fixed $x \in \mathcal{X}$, we have*

$$
\mathrm{KL}\big(\mathbb{P}(\cdot, \cdot \,|\, x, \theta) \| \mathbb{P}(\cdot, \cdot \,|\, x, \theta')\big) = A(x, \theta') - A(x, \theta) + \langle \nabla A(x, \theta), f_\theta - f_{\theta'} \rangle.
$$

*Proof of Lemma D.1.* By the definition of KL-divergence, we have

$$
\begin{aligned}
\mathrm{KL}\big(\mathbb{P}(\cdot, \cdot \,|\, x, \theta) \| \mathbb{P}(\cdot, \cdot \,|\, x, \theta')\big) &= \sum_{(z,y) \in \mathcal{Z} \times \mathcal{Y}} \mathbb{P}(z, y \,|\, x, \theta) \cdot \log \frac{\mathbb{P}(z, y \,|\, x, \theta)}{\mathbb{P}(z, y \,|\, x, \theta')} \\
&= \sum_{(z,y) \in \mathcal{Z} \times \mathcal{Y}} \mathbb{P}(z, y \,|\, x, \theta) \cdot \Big[ f_\theta(x, z, y) - f_{\theta'}(x, z, y) + A(x, \theta') - A(x, \theta) \Big] \\
&= A(x, \theta') - A(x, \theta) + \big\langle \mathbb{E}_{(z,y) \sim \mathbb{P}(\cdot, \cdot \,|\, x, \theta)}[\mathcal{K}((x, z, y), \cdot)], f_\theta - f_{\theta'} \big\rangle,
\end{aligned}
\tag{D.1}
$$

where the last equality uses (i) $\sum_{(z,y) \in \mathcal{Z} \times \mathcal{Y}} \mathbb{P}(z, y \,|\, x, \theta) = 1$; and (ii) the assumption that $f_\theta \in \mathcal{H}$ (Assumption 3.2). Meanwhile, by Assumption 3.2, we have

$$
A(x, \theta) = \log \sum_{(z,y) \in \mathcal{Z} \times \mathcal{Y}} \exp(f_\theta(x, z, y)) = \log \sum_{(z,y) \in \mathcal{Z} \times \mathcal{Y}} \exp(\langle \mathcal{K}((x, z, y), \cdot), f_\theta \rangle,
$$

which implies that

$$
\begin{aligned}
\nabla A(x, \theta) &= \frac{\nabla \big( \sum_{(z,y) \in \mathcal{Z} \times \mathcal{Y}} \exp(\langle \mathcal{K}((x, z, y), \cdot), f_\theta \rangle) \big)}{\sum_{(z,y) \in \mathcal{Z} \times \mathcal{Y}} \exp(\langle \mathcal{K}((x, z, y), \cdot), f_\theta \rangle)} \\
&= \sum_{(z,y) \in \mathcal{Z} \times \mathcal{Y}} \frac{\exp(\langle \mathcal{K}((x, z, y), \cdot), f_\theta \rangle)}{\sum_{(z,y) \in \mathcal{Z} \times \mathcal{Y}} \exp(\langle \mathcal{K}((x, z, y), \cdot), f_\theta \rangle)} \cdot \mathcal{K}((x, z, y), \cdot) \\
&= \sum_{(z,y) \in \mathcal{Z} \times \mathcal{Y}} \mathbb{P}(z, y \,|\, x, \theta) \cdot \mathcal{K}((x, z, y), \cdot),
\end{aligned}
\tag{D.2}
$$

where $\nabla$ is a functional gradient. Combining (D.1) and (D.2), we have

$$
\mathrm{KL}\big(\mathbb{P}(z, y \,|\, x, \theta) \| \mathbb{P}(z, y \,|\, x, \theta')\big) = A(x, \theta') - A(x, \theta) + \langle \nabla A(x, \theta), f_\theta - f_{\theta'} \rangle,
$$

which finishes the proof of Lemma D.1.  □

**Lemma D.2.** *It holds that*

$$\nabla \log \mathbb{P}(z \in \mathscr{Z}, y \in \mathscr{Y}, o \in \mathscr{O} \,|\, x, \theta) = \mathbb{E}_{(z,y) \sim \mathbb{Q}(\cdot, \cdot \,|\, x, \theta)}[\mathcal{K}((x, z, y), \cdot)] - \nabla A(x, \theta),$$

*where* $\mathbb{Q}(z, y \,|\, x) = \frac{\sum_{o \in \mathscr{O}} \mathbb{P}(z, y, o \,|\, x, \theta)}{\sum_{(z,y,o) \in \mathscr{Z} \times \mathscr{Y} \times \mathscr{O}} \mathbb{P}(z, y, o \,|\, x, \theta)}.$

*Proof.* By definition, we have

$$\begin{aligned}
\nabla \log \mathbb{P}(z \in \mathscr{Z}, y \in \mathscr{Y}, o \in \mathscr{O} \,|\, x, \theta) &= \nabla \log \sum_{(z,y,o) \in \mathscr{Z} \times \mathscr{Y} \times \mathscr{O}} \mathbb{P}(z, y, o \,|\, x, \theta) \\
&= \frac{\sum_{(z,y,o) \in \mathscr{Z} \times \mathscr{Y} \times \mathscr{O}} \nabla \mathbb{P}(z, y, o \,|\, x, \theta)}{\sum_{(z,y,o) \in \mathscr{Z} \times \mathscr{Y} \times \mathscr{O}} \mathbb{P}(z, y, o \,|\, x, \theta)} \\
&= \frac{\sum_{(z,y,o) \in \mathscr{Z} \times \mathscr{Y} \times \mathscr{O}} \left( \mathbb{P}(o \,|\, x, z, y) \cdot \nabla \mathbb{P}(z, y \,|\, x, \theta) \right)}{\sum_{(z,y,o) \in \mathscr{Z} \times \mathscr{Y} \times \mathscr{O}} \mathbb{P}(z, y, o \,|\, x, \theta)},
\end{aligned} \tag{D.3}$$

where the last equality follows from (3.1). Meanwhile, by Assumption 3.2, we further have

$$\begin{aligned}
\nabla \mathbb{P}(z, y \,|\, x, \theta) &= \nabla \exp(f_\theta(x, z, y) - A(x, \theta)) \\
&= \exp(f_\theta(x, z, y) - A(x, \theta)) \cdot [\nabla \langle \mathcal{K}((x, z, y), \cdot), f_\theta \rangle - \nabla A(x, \theta)] \\
&= \mathbb{P}(z, y \,|\, x, \theta) \cdot [\mathcal{K}((x, z, y), \cdot) - \nabla A(x, \theta)].
\end{aligned} \tag{D.4}$$

Combining (D.3) and (D.4), we have

$$\begin{aligned}
\nabla \log \mathbb{P}(z \in \mathscr{Z}, y \in \mathscr{Y}, o \in \mathscr{O} \,|\, x, \theta) &= \frac{\sum_{(z,y,o) \in \mathscr{Z} \times \mathscr{Y} \times \mathscr{O}} \left( \mathbb{P}(o \,|\, x, z, y) \cdot \nabla \mathbb{P}(z, y \,|\, x, \theta) \right)}{\sum_{(z,y,o) \in \mathscr{Z} \times \mathscr{Y} \times \mathscr{O}} \mathbb{P}(z, y, o \,|\, x, \theta)} \\
&= \sum_{(z,y) \in \mathscr{Z} \times \mathscr{Y}} \frac{\sum_{o \in \mathscr{O}} \mathbb{P}(z, y, o \,|\, x, \theta)}{\sum_{(z,y,o) \in \mathscr{Z} \times \mathscr{Y} \times \mathscr{O}} \mathbb{P}(z, y, o \,|\, x, \theta)} \cdot [\mathcal{K}((x, z, y), \cdot) - \nabla A(x, \theta)] \\
&= \mathbb{E}_{(z,y) \sim \mathbb{Q}(\cdot, \cdot \,|\, x, \theta)}[\mathcal{K}((x, z, y), \cdot)] - \nabla A(x, \theta).
\end{aligned} \tag{D.5}$$

This finishes the proof of Lemma D.2. □

Now we start the proof of Theorem 3.3.

*Proof of Theorem 3.3.* By the updating rule of $\theta_{t+1}$ in (3.5), we have

$$\begin{aligned}
\theta_{t+1} &= \operatorname*{argmax}_\theta \left\{ \sum_{(z,y,o) \in \mathscr{Z} \times \mathscr{Y}} \log \mathbb{P}(z, y \,|\, x, \theta) \cdot \mathbb{Q}(z, y, o \,|\, x, \theta_t) \right\} \\
&= \operatorname*{argmax}_\theta \left\{ \sum_{(z,y) \in \mathscr{Z} \times \mathscr{Y}} \log \mathbb{P}(z, y \,|\, x, \theta) \cdot \mathbb{Q}(z, y \,|\, x, \theta_t) \right\},
\end{aligned}$$

where we use the notation $\mathbb{Q}(z, y \,|\, x, \theta_t) = \sum_{o \in \mathscr{O}} \mathbb{Q}(z, y, o \,|\, x, \theta_t)$. Furthermore, we have

$$\begin{aligned}
\theta_{t+1} &= \operatorname*{argmax}_\theta \left\{ \sum_{(z,y) \in \mathscr{Z} \times \mathscr{Y}} \log \mathbb{P}(z, y \,|\, x, \theta) \cdot \mathbb{Q}(z, y \,|\, x, \theta_t) \right\} \\
&= \operatorname*{argmax}_\theta \left\{ \sum_{(z,y) \in \mathscr{Z} \times \mathscr{Y}} \log \frac{\mathbb{P}(z, y \,|\, x, \theta)}{\mathbb{P}(z, y \,|\, x, \theta_t)} \cdot \mathbb{Q}(z, y \,|\, x, \theta_t) \right\} \\
&= \operatorname*{argmax}_\theta \left\{ \left\langle \mathbb{E}_{(z,y) \sim \mathbb{Q}(\cdot, \cdot \,|\, x, \theta_t)}[\mathcal{K}((x, z, y), \cdot)], f_\theta - f_{\theta_t} \right\rangle + A(x, \theta_t) - A(x, \theta) \right\},
\end{aligned} \tag{D.6}$$

where the last equality is implied by

$$\log \frac{\mathbb{P}(z, y \,|\, x, \theta)}{\mathbb{P}(z, y \,|\, x, \theta_t)} = f_\theta(x, z, y) - f_{\theta_t}(x, z, y) + A(x, \theta_t) - A(x, \theta)$$
$$= \langle \mathcal{K}((x, z, y), \cdot), f_\theta - f_{\theta_t} \rangle + A(x, \theta_t) - A(x, \theta).$$

Combining (D.6) and Lemma D.2, we have

$$\theta_{t+1} = \underset{\theta}{\operatorname{argmax}} \Big\{ \langle \nabla \log \mathbb{P}(z \in \mathcal{Z}, y \in \mathcal{Y}, o \in \mathcal{O} \,|\, x, \theta_t) + \nabla A(x, \theta_t), f_\theta - f_{\theta_t} \rangle + A(x, \theta_t) - A(x, \theta) \Big\}. \qquad \text{(D.7)}$$

By the optimality condition, this implies that

$$\nabla A(\theta_{t+1}) - \nabla A(\theta_t) = \nabla \log \mathbb{P}(z \in \mathcal{Z}, y \in \mathcal{Y}, o \in \mathcal{O} \,|\, x, \theta_t). \qquad \text{(D.8)}$$

We have

$$\langle \nabla A(x, \theta_{t+1}) - \nabla A(x, \theta_t), f_{\theta^*} - f_{\theta_{t+1}} \rangle$$
$$= \langle \nabla A(x, \theta_t), f_{\theta_{t+1}} - f_{\theta_t} \rangle - A(x, \theta_{t+1}) + A(x, \theta_t)$$
$$\quad - \langle \nabla A(x, \theta_t), f_{\theta^*} - f_{\theta_t} \rangle + A(x, \theta^*) - A(x, \theta_t)$$
$$\quad + \langle \nabla A(x, \theta_{t+1}), f_{\theta^*} - f_{\theta_{t+1}} \rangle - A(x, \theta^*) + A(x, \theta_{t+1})$$
$$= -\mathrm{KL}\big(\mathbb{P}(\cdot, \cdot \,|\, x, \theta_t) \| \mathbb{P}(\cdot, \cdot \,|\, x, \theta_{t+1})\big) + \mathrm{KL}\big(\mathbb{P}(\cdot, \cdot \,|\, x, \theta_t) \| \mathbb{P}(\cdot, \cdot \,|\, x, \theta^*)\big) - \mathrm{KL}\big(\mathbb{P}(\cdot, \cdot \,|\, x, \theta_{t+1}) \| \mathbb{P}(\cdot, \cdot \,|\, x, \theta^*)\big),$$

where the last equality follows from Lemma D.1. This is equivalent to

$$\mathrm{KL}\big(\mathbb{P}(\cdot, \cdot \,|\, x, \theta_t) \| \mathbb{P}(\cdot, \cdot \,|\, x, \theta^*)\big) - \mathrm{KL}\big(\mathbb{P}(\cdot, \cdot \,|\, x, \theta_{t+1}) \| \mathbb{P}(\cdot, \cdot \,|\, x, \theta^*)\big)$$
$$= \langle \nabla A(\theta_{t+1}) - \nabla A(\theta_t), f_{\theta^*} - f_{\theta_{t+1}} \rangle + \mathrm{KL}\big(\mathbb{P}(\cdot, \cdot \,|\, x, \theta_t) \| \mathbb{P}(\cdot, \cdot \,|\, x, \theta_{t+1})\big)$$
$$= \underbrace{\langle \nabla \log \mathbb{P}(z \in \mathcal{Z}, y \in \mathcal{Y}, o \in \mathcal{O} \,|\, x, \theta_t), f_{\theta^*} - f_{\theta_{t+1}} \rangle}_{(\star)} + \mathrm{KL}\big(\mathbb{P}(\cdot, \cdot \,|\, x, \theta_t) \| \mathbb{P}(\cdot, \cdot \,|\, x, \theta_{t+1})\big), \qquad \text{(D.9)}$$

where the last equality follows from (D.8). For the Term $(\star)$, we have

$$(\star) = \langle \nabla \log \mathbb{P}(z \in \mathcal{Z}, y \in \mathcal{Y}, o \in \mathcal{O} \,|\, x, \theta_t), f_{\theta^*} - f_{\theta_{t+1}} \rangle$$
$$= \underbrace{\langle \nabla \log \mathbb{P}(z \in \mathcal{Z}, y \in \mathcal{Y}, o \in \mathcal{O} \,|\, x, \theta_t), f_{\theta^*} - f_{\theta_t} \rangle}_{\text{(I)}} + \underbrace{\langle \nabla \log \mathbb{P}(z \in \mathcal{Z}, y \in \mathcal{Y}, o \in \mathcal{O} \,|\, x, \theta_t), f_{\theta_t} - f_{\theta_{t+1}} \rangle}_{\text{(II)}}.$$
$$\text{(D.10)}$$

**Term (I).** By the local concave assumption, we have

$$\text{Term (I) in (D.10)} \geq \log \mathbb{P}(z \in \mathcal{Z}, y \in \mathcal{Y}, o \in \mathcal{O} \,|\, x, \theta^*) - \log \mathbb{P}(z \in \mathcal{Z}, y \in \mathcal{Y}, o \in \mathcal{O} \,|\, x, \theta_t). \qquad \text{(D.11)}$$

**Term (II).** By Lemma 3.1, we have

$$\log \mathbb{P}(z \in \mathcal{Z}, y \in \mathcal{Y}, o \in \mathcal{O} \,|\, x, \theta_{t+1})$$
$$= \sum_{(z, y, o) \in \mathcal{Z} \times \mathcal{Y} \times \mathcal{O}} [\log \mathbb{P}(z, y, o \,|\, x, \theta_{t+1}) - \log \mathbb{Q}(z, y, o \,|\, x, \theta_{t+1})] \cdot \mathbb{Q}(z, y, o \,|\, x, \theta_t), \qquad \text{(D.12)}$$

where

$$\mathbb{Q}(z, y, o \,|\, x, \theta) = \frac{\mathbb{P}(z, y, o \,|\, x, \theta)}{\sum_{(z', y', o') \in \mathcal{Z} \times \mathcal{Y} \times \mathcal{O}} \mathbb{P}(z', y', o' \,|\, x, \theta)} = \frac{\mathbb{P}(z, y, o \,|\, x, \theta)}{\mathbb{P}(z \in \mathcal{Z}, y \in \mathcal{Y}, o \in \mathcal{O} \,|\, x, \theta)}. \qquad \text{(D.13)}$$

Note that $\mathbb{Q}(\cdot,\cdot,\cdot\,|\,x,\theta_t)$ is a distribution over $\mathscr{L}\times\mathscr{Y}\times\mathscr{O}$, which means that $\sum_{(z,y,o)\in\mathscr{L}\times\mathscr{Y}\times\mathscr{O}}\mathbb{Q}(z,y,o\,|\,x,\theta_t)=1$. Hence, we have

$$\log\mathbb{P}(z\in\mathscr{L},y\in\mathscr{Y},o\in\mathscr{O}\,|\,x,\theta_t)=\sum_{(z,y,o)\in\mathscr{L}\times\mathscr{Y}\times\mathscr{O}}\mathbb{Q}(z,y,o\,|\,x,\theta_t)\cdot\log\mathbb{P}(z\in\mathscr{L},y\in\mathscr{Y},o\in\mathscr{O}\,|\,x,\theta_t)$$

$$=\sum_{(z,y,o)\in\mathscr{L}\times\mathscr{Y}\times\mathscr{O}}\mathbb{Q}(z,y,o\,|\,x,\theta_t)\cdot\log\frac{\mathbb{P}(z,y,o\,|\,x,\theta_t)}{\mathbb{Q}(z,y,o\,|\,x,\theta_t)}, \tag{D.14}$$

where the last equality is implied by the definition of $\mathbb{Q}(z,y,o\,|\,x,\theta)$ in (D.13). Combining (D.12) and (D.14), we have

$$\log\mathbb{P}(z\in\mathscr{L},y\in\mathscr{Y},o\in\mathscr{O}\,|\,x,\theta_{t+1})-\log\mathbb{P}(z\in\mathscr{L},y\in\mathscr{Y},o\in\mathscr{O}\,|\,x,\theta_t)$$

$$=\sum_{(z,y,o)\in\mathscr{L}\times\mathscr{Y}\times\mathscr{O}}\mathbb{Q}(z,y,o\,|\,x,\theta_t)\cdot\log\frac{\mathbb{P}(z,y,o\,|\,x,\theta_{t+1})}{\mathbb{P}(z,y,o\,|\,x,\theta_t)}+\mathrm{KL}\big(\mathbb{Q}(\cdot,\cdot,\cdot\,|\,x,\theta_t)\|\mathbb{Q}(\cdot,\cdot,\cdot\,|\,x,\theta_{t+1})\big)$$

$$\geq\sum_{(z,y,o)\in\mathscr{L}\times\mathscr{Y}\times\mathscr{O}}\mathbb{Q}(z,y,o\,|\,x,\theta_t)\cdot\log\frac{\mathbb{P}(z,y,o\,|\,x,\theta_{t+1})}{\mathbb{P}(z,y,o\,|\,x,\theta_t)}$$

$$=\sum_{(z,y,o)\in\mathscr{L}\times\mathscr{Y}\times\mathscr{O}}\mathbb{Q}(z,y,o\,|\,x,\theta_t)\cdot\log\frac{\mathbb{P}(z,y\,|\,x,\theta_{t+1})}{\mathbb{P}(z,y\,|\,x,\theta_t)}, \tag{D.15}$$

where the third equality follows from the non-negativity of KL-divergence, and the last equality is implied by (3.1). By Assumption 3.2, we can rewrite the right-handside of (D.15) as

$$\sum_{(z,y,o)\in\mathscr{L}\times\mathscr{Y}}\mathbb{Q}(z,y,o\,|\,x,\theta_t)\cdot\big(f_{\theta_{t+1}}(x,z,y)-f_{\theta_t}(x,z,y)\big)+A(x,\theta_t)-A(x,\theta_{t+1})$$

$$=\big\langle\mathbb{E}_{(z,y)\sim\mathbb{Q}(\cdot,\cdot\,|\,x,\theta_t)}[\mathcal{K}((x,z,y),\cdot)],f_{\theta_{t+1}}-f_{\theta_t}\big\rangle+A(x,\theta_t)-A(x,\theta_{t+1})$$

$$=\big\langle\mathbb{E}_{(z,y)\sim\mathbb{Q}(\cdot,\cdot\,|\,x,\theta_t)}[\mathcal{K}((x,z,y),\cdot)]-\nabla A(x,\theta_t),f_{\theta_{t+1}}-f_{\theta_t}\big\rangle-\mathrm{KL}\big(\mathbb{P}(\cdot,\cdot\,|\,x,\theta_t)\|\mathbb{P}(\cdot,\cdot\,|\,x,\theta_{t+1})\big)$$

$$=\big\langle\nabla\log\mathbb{P}(z\in\mathscr{L},y\in\mathscr{Y},o\in\mathscr{O}\,|\,x,\theta_t),f_{\theta_{t+1}}-f_{\theta_t}\big\rangle-\mathrm{KL}\big(\mathbb{P}(\cdot,\cdot\,|\,x,\theta_t)\|\mathbb{P}(\cdot,\cdot\,|\,x,\theta_{t+1})\big), \tag{D.16}$$

where the third line is implied by Lemma D.1, and the last equality follows from (D.5). Plugging (D.16) into (D.15), we have

$$\text{Term (II) in (D.10)}=\big\langle\nabla\log\mathbb{P}(z\in\mathscr{L},y\in\mathscr{Y},o\in\mathscr{O}\,|\,x,\theta_t),f_{\theta_t}-f_{\theta_{t+1}}\big\rangle$$

$$\geq\log\mathbb{P}(z\in\mathscr{L},y\in\mathscr{Y},o\in\mathscr{O}\,|\,x,\theta_t)-\log\mathbb{P}(z\in\mathscr{L},y\in\mathscr{Y},o\in\mathscr{O}\,|\,x,\theta_{t+1})$$

$$-\mathrm{KL}\big(\mathbb{P}(\cdot,\cdot\,|\,x,\theta_t)\|\mathbb{P}(\cdot,\cdot\,|\,x,\theta_{t+1})\big). \tag{D.17}$$

Combining (D.10), (D.11), and (D.17), we have

$$(\star)\geq\text{Term (I)}+\text{Term (II)}$$

$$\geq\log\mathbb{P}(z\in\mathscr{L},y\in\mathscr{Y},o\in\mathscr{O}\,|\,x,\theta^*)-\log\mathbb{P}(z\in\mathscr{L},y\in\mathscr{Y},o\in\mathscr{O}\,|\,x,\theta_t)+\log\mathbb{P}(z\in\mathscr{L},y\in\mathscr{Y},o\in\mathscr{O}\,|\,x,\theta_t)$$

$$-\log\mathbb{P}(z\in\mathscr{L},y\in\mathscr{Y},o\in\mathscr{O}\,|\,x,\theta_{t+1})-\mathrm{KL}\big(\mathbb{P}(\cdot,\cdot\,|\,x,\theta_t)\|\mathbb{P}(\cdot,\cdot\,|\,x,\theta_{t+1})\big)$$

$$=\log\mathbb{P}(z\in\mathscr{L},y\in\mathscr{Y},o\in\mathscr{O}\,|\,x,\theta^*)-\log\mathbb{P}(z\in\mathscr{L},y\in\mathscr{Y},o\in\mathscr{O}\,|\,x,\theta_{t+1})$$

$$-\mathrm{KL}\big(\mathbb{P}(\cdot,\cdot\,|\,x,\theta_t)\|\mathbb{P}(\cdot,\cdot\,|\,x,\theta_{t+1})\big). \tag{D.18}$$

Putting (D.9) and (D.18) together, we obtain

$$\mathrm{KL}\big(\mathbb{P}(\cdot,\cdot\,|\,x,\theta_t)\|\mathbb{P}(\cdot,\cdot\,|\,x,\theta^*)\big)-\mathrm{KL}\big(\mathbb{P}(\cdot,\cdot\,|\,x,\theta_{t+1})\|\mathbb{P}(\cdot,\cdot\,|\,x,\theta^*)\big)$$

$$\geq\log\mathbb{P}(z\in\mathscr{L},y\in\mathscr{Y},o\in\mathscr{O}\,|\,x,\theta^*)-\log\mathbb{P}(z\in\mathscr{L},y\in\mathscr{Y},o\in\mathscr{O}\,|\,x,\theta_{t+1}).$$

Telescoping this inequality from 0 to $T-1$, we obtain that which implies that

$$\min_{1\leq t\leq T}\{\log\mathbb{P}(z\in\mathscr{L},y\in\mathscr{Y},o\in\mathscr{O}\,|\,x,\theta^*)-\log\mathbb{P}(z\in\mathscr{L},y\in\mathscr{Y},o\in\mathscr{O}\,|\,x,\theta_t)\}\leq\frac{\mathrm{KL}\big(\mathbb{P}(\cdot,\cdot\,|\,x,\theta_1)\|\mathbb{P}(\cdot,\cdot\,|\,x,\theta^*)\big)}{T},$$

which finishes the proof of Theorem 3.3. $\square$

## D.2. Additional Theoretical Results

**Theorem D.3.** *Suppose Assumption 3.2 holds. Then we have*

$$\min_{1 \le t \le T} \mathrm{KL}\big(\mathbb{P}(\cdot, \cdot \,|\, x, \theta_{t+1}) \| \mathbb{P}(\cdot, \cdot \,|\, x, \theta_t)\big) \le \frac{\log \mathbb{P}(z \in \mathscr{Z}, y \in \mathscr{Y}, o \in \mathscr{O} \,|\, x, \theta_{T+1}) - \log \mathbb{P}(z \in \mathscr{Z}, y \in \mathscr{Y}, o \in \mathscr{O} \,|\, x, \theta_1)}{T}.$$

*Proof of Theorem D.3.* By the same derivation of (D.17), we have

$$\begin{aligned}
\mathrm{KL}\big(\mathbb{P}(\cdot, \cdot \,|\, x, \theta_t) \| \mathbb{P}(\cdot, \cdot \,|\, x, \theta_{t+1})\big) &\ge \log \mathbb{P}(z \in \mathscr{Z}, y \in \mathscr{Y}, o \in \mathscr{O} \,|\, x, \theta_t) - \log \mathbb{P}(z \in \mathscr{Z}, y \in \mathscr{Y}, o \in \mathscr{O} \,|\, x, \theta_{t+1}) \\
&\quad - \big\langle \nabla \log \mathbb{P}(z \in \mathscr{Z}, y \in \mathscr{Y}, o \in \mathscr{O} \,|\, x, \theta_t), f_{\theta_t} - f_{\theta_{t+1}} \big\rangle \\
&= \log \mathbb{P}(z \in \mathscr{Z}, y \in \mathscr{Y}, o \in \mathscr{O} \,|\, x, \theta_t) - \log \mathbb{P}(z \in \mathscr{Z}, y \in \mathscr{Y}, o \in \mathscr{O} \,|\, x, \theta_{t+1}) \\
&\quad - \big\langle \nabla A(\theta_{t+1}) - \nabla A(\theta_t), f_{\theta_t} - f_{\theta_{t+1}} \big\rangle,
\end{aligned} \tag{D.19}$$

where the last equality is implied by the optimality condition in (D.8). By Lemma D.1, we have

$$\mathrm{KL}\big(\mathbb{P}(\cdot, \cdot \,|\, x, \theta_t) \| \mathbb{P}(\cdot, \cdot \,|\, x, \theta_{t+1})\big) = A(x, \theta_{t+1}) - A(x, \theta_t) + \big\langle \nabla A(x, \theta_t), f_{\theta_t} - f_{\theta_{t+1}} \big\rangle. \tag{D.20}$$

Combining (D.19) and (D.20), we have

$$\begin{aligned}
A(x, \theta_{t+1}) - A(x, \theta_t) &+ \big\langle \nabla A(\theta_{t+1}), f_{\theta_t} - f_{\theta_{t+1}} \big\rangle \\
&\ge \log \mathbb{P}(z \in \mathscr{Z}, y \in \mathscr{Y}, o \in \mathscr{O} \,|\, x, \theta_t) - \log \mathbb{P}(z \in \mathscr{Z}, y \in \mathscr{Y}, o \in \mathscr{O} \,|\, x, \theta_{t+1}).
\end{aligned}$$

Together with Lemma D.1, we have

$$\begin{aligned}
\log \mathbb{P}(z \in \mathscr{Z}, y \in \mathscr{Y}, o \in \mathscr{O} \,|\, x, \theta_{t+1}) &- \log \mathbb{P}(z \in \mathscr{Z}, y \in \mathscr{Y}, o \in \mathscr{O} \,|\, x, \theta_t) \\
&\ge A(x, \theta_t) - A(x, \theta_{t+1}) + \big\langle \nabla A(\theta_{t+1}), f_{\theta_{t+1}} - f_{\theta_t} \big\rangle = \mathrm{KL}\big(\mathbb{P}(\cdot, \cdot \,|\, x, \theta_{t+1}) \| \mathbb{P}(\cdot, \cdot \,|\, x, \theta_t)\big).
\end{aligned}$$

Telescoping this inequality across $t \in [T]$, we have

$$\begin{aligned}
\min_{t \in [T]} \big\{ \mathrm{KL}\big(\mathbb{P}(\cdot, \cdot \,|\, x, \theta_{t+1}) \| \mathbb{P}(\cdot, \cdot \,|\, x, \theta_t)\big) \big\} &\le \frac{\sum_{t=1}^T \mathrm{KL}\big(\mathbb{P}(\cdot, \cdot \,|\, x, \theta_{t+1}) \| \mathbb{P}(\cdot, \cdot \,|\, x, \theta_t)\big)}{T} \\
&\le \frac{\log \mathbb{P}(z \in \mathscr{Z}, y \in \mathscr{Y}, o \in \mathscr{O} \,|\, x, \theta_{T+1}) - \log \mathbb{P}(z \in \mathscr{Z}, y \in \mathscr{Y}, o \in \mathscr{O} \,|\, x, \theta_1)}{T},
\end{aligned}$$

which finishes the proof of Theorem D.3. $\qquad\square$

# E. Implementation Details

## E.1. Implementation Details for Math Task

**Implementation of BRiTE.** The BRiTE algorithm is run on 4 NVIDIA H100 during training. We leverage the PPO pipeline (Schulman et al., 2017) to learn the sampling policy $\mathbb{Q}$ (3.4) with a learning rate of $5e-7$ and a batch size of 1. For the subsequent SFT on rationales sampled by $\mathbb{Q}$, we set the learning rate to $5e-5$ and the batch size to 2. We adopt the LoRA (Hu et al., 2021) training for both steps, where the `r` is set to 32 and `lora alpha` is set to 128.

**Implementation of RS.** For rejection sampling, we set the temperature of the model's generation to $1.0$, sample $N = 30$ candidate rationales for each problem $x$, and select the best rationales $z_{\mathrm{rs}}$. we filter generations by selecting those that produce the correct final answer. The model is then fine-tuned on these problem-rationale-answer tuples $\{x, z_{\mathrm{rs}}, y^*\}$ using the same learning rate and LoRA parameters as in BRiTE.

**Implementation of SFT.** For SFT on original datasets with human-annotated rationales, we use the same LoRA parameters as in BRiTE. The learning rates for `Mistral-7B-Instruct-v0.2` and `Gemma-1.1-7B-it` are set to $5e-5$. For `Gemma-2-9b-it`, the learning rate is $5e-6$, and for `Llama-3-8B-Instruct`, it is $8e-7$. These smaller learning rates are chosen to mitigate overfitting and ensure optimal performance.

**Implementation of RLHF Algorithms.** For both BRiTE DPO and iterative DPO, we select 6k data points per iteration from the `RLHF4MATH/prompt_iter1,2,3` dataset[3] over three iterations. The learning rates for DPO training are configured as follows: `Mistral-7B-Instruct-v0.2` uses a learning rate of $2e-7$, `Gemma-1.1-7B-it` employs $4e-7$, while `Gemma-2-9B-it` and `Llama-3-8B-Instruct` are set to $5e-7$. The generation temperature is set to 1.0, and each prompt sample $N = 30$ responses.

**Evaluation.** We utilize the evaluation function provided in the `Qwen2.5-Math` repository[4] to assess the models' performance on the test sets of GSM8K and MATH.

### E.2. Implementation Details for Coding Generation Tasks

**Datasets.** For the code generation task, We choose the first 4000 rows of the `educational instruct` split of the dataset `OpenCoder-LLM/opc-sft-stage2`[5](Huang et al., 2024) as the training dataset, which comprises (instruction, code, test case) tuples. The entire dataset is generated with an algorithmic corpus as a seed and validated through a Python compiler.

**Models.** We choose the language model `deepseek-ai/deepseek-coder-6.7b-instruct`[6] with around seven billion active parameters as the base model, which is especially pretrained and fined-tuned on the code corpus for better code generation performance.

**Baselines.** Similar to the mathematics tasks, we use Reject Sampling (RS) and SFT as the baselines. We follow a rejection sampling approach similar to prior math section. The model generates candidate rationales with a temperature of 1.0, sample $N = 30$ rationales for problem $x$. The best rationales, $z_{rs}$, is then selected. The model is then fine-tuned on these rationale-answer tuples $\{x, z_{rs}, y\}$. Notably, unlike the mathematical setting, filtering correct generations in the code generation task requires candidates to pass all unit tests and receive positive compiler feedback.

**Benchmarks.** We evaluate the trained models on the HumanEval task from Evalplus (Liu et al., 2023) and the instruct split of the BigCodeBench (Zhuo et al., 2024). These popular benchmarks evaluate how the language models complete the partial code snippet and generate a full code snippet according to natural language instructions.

**Training Details.** We use 4 NVIDIA A100 GPUs for all the training. We leverage the PPO pipeline (Schulman et al., 2017) to learn the sampling policy $\mathbb{Q}$ (3.4) with a learning rate of $5e-7$ and a batch size of 1. With the rationales sampled from $\mathbb{Q}$ of BRiTE, we set the learning rate as $5e-5$ and the batch size as 2 in the SFT stage of BRiTE. For the baselines, we also set the learning rate as $5e-5$ and the batch size as 2 in the SFT algorithm. For the fine-tune stage of reject sampling, we use the same learning rate as in BRiTE. To save the computation budget, we adopt the LoRA (Hu et al., 2021) training for all the optimization phases, where the $r$ is set to 32 and `lora alpha` is set to 128.

**Evaluation Details and Results.** We report the pass@1 scores (the success rate of the first attempt) with the greedy decoding for the base model, SFT algorithm, Reject Sampling (RS) algorithm, and our proposed algorithm BRiTE-SFT in Table 2. Results show that our proposed algorithm BRiTE can help LLMs improve their code generation ability and demonstrate the performance gain of BRiTE compared with the baselines. We also remark that rejection sampling-based algorithms require the code datasets to be equipped with correct unit tests, which are unnecessary for BRiTE.

---

[3] https://huggingface.co/RLHF4MATH
[4] https://github.com/QwenLM/Qwen2.5-Math
[5] https://huggingface.co/datasets/OpenCoder-LLM/opc-sft-stage2/tree/main
[6] https://huggingface.co/deepseek-ai/deepseek-coder-6.7b-instruct

