# OpenReview forum: "BRiTE: Bootstrapping Reinforced Thinking Process to Enhance Language Model Reasoning"
_ICML.cc/2025/Conference — ICML 2025 poster_

### Official Review · Reviewer_8vbs · 2025-02-27

**Overall Recommendation:** 3

**Summary:**

Authors propose a new approach to incorporate latent thinking processes and evaluation signals in the model training. Authors propose two-step approach: 1. generate high-quality rationales by approximating desired posterior of thought given the question-answer pair; this approach relies on reinforcement learning, using a novel token-level reward function; and 2. enhance the base LLM by maximizing the joint probability of rationale generation with respect to the model’s parameters. Importantly, this approach does not require reference chain-of-thought data, but instead relies on model to generate rationales. Authors provide extensive theoretical background for the proposed approach. Experiments on math and code generation tasks show consistent improvements across five LLM, including Gemma, Llama, Mistral, and DeepSeek sizes ranging from 1B to 8B.

## update after rebattle
I read through other reviews and responses. I think the paper has merits and leaning towards acceptance.

**Claims And Evidence:**

Yes. Authors propose a new method to enhance reasoning abilities of the models, provide extensive theoretical investigations and compare performance of five models on four tasks showcasing consistent improvements.

**Essential References Not Discussed:**

N/A

**Experimental Designs Or Analyses:**

Yes. I think authors included solid variety of the models to show the generalization ability of their approach. However, I would highlight a few issues:
1. Authors test their approach on math and code reasoning tasks. It would be interesting to see how this approach will work on other types of reasoning tasks (for ex, some popular benchmarks include MMLU, GPQA, BIG-Bench Hard, etc).
2. Proposed approach for BRiTE-SFT involves two stages of training, while traditional SFT methods authors compare with involve 2 stage of training. Authors to not provide any theoretical or actual cost estimates required to train with BRiTE. Given this process is two-staged, I assume it would use more GPU days and - depending on how much more - might not be directly comparable with direct SFT.

**Methods And Evaluation Criteria:**

Yes.

**Other Comments Or Suggestions:**

Page 3, line 113: Initially, an initial state  -> An initial state
Page 3, line 126: .. l framework fro LLM reason and unified ... - not sure what you mean here

**Other Strengths And Weaknesses:**

This work is novel and provides extensive theoretical and practical details. In my opinion, the main weakness of the paper lies in its experimental part. Proposed approach for BRiTE-SFT involves two stages of training, while traditional SFT methods authors compare with involve 2 stage of training. Authors to not provide any theoretical or actual cost estimates required to train with BRiTE. Given this process is two-staged, I assume it would use more GPU days and - depending on how much more - might not be directly comparable with direct SFT.

**Questions For Authors:**

1. p. 7 line 383: How do you select best and worse response based on correctness of y? Given 30 CoT, k will have correct answer, and (30-k) incorrect. How do you select best of k and worst of (30-k)? If it is not mentioned in the paper, please add elaboration.

2. Please include theoretical or actual cost estimates required to train with BRiTE-SFT, and add comparison with traditional SFT you used for Table 1.

**Relation To Broader Scientific Literature:**

This work further advances popular RL-based methods used for LLM post-training.

**Theoretical Claims:**

I checked the correctness of proofs and theoretical claims, however I'm not completely confident in my evaluations.

---

> ### Author Rebuttal · Authors · 2025-03-31
>
> **Comments on "Experimental Designs Or Analyses":**  Authors test their approach on math and code reasoning tasks. It would be interesting to see how this approach will work on other types of reasoning tasks (for ex, some popular benchmarks include MMLU, GPQA, BIG-Bench Hard, etc).
>
> **Response:** We perform BRiTE on the model *Qwen/Qwen2.5-7B* with a mixed dataset (the size is around 40K) of *RUC-AIBOX/STILL-3-Preview-RL-Data*, MATH, and historical AIME problems (excluding AIME 2024).  We evaluate models trained using RS baseline and BRiTE on a diverse set of reasoning benchmarks, including Math-500, Minerva Math, Olympiad Bench, AIME24, AMC23, and GPQA Diamond. The pass@1 accuracy (64 sampling times for AIME24 and AMC23 and 8 sampling times for the others) for each benchmark is reported in the table below.
> |    Method           | Math500 | Minerva_math | Olympiadbench | Aime24 | Amc23 | Gpqa_diamond |
> |----------------|---------|--------------|---------------|-----------|----------|--------------|
> | Base Model          | 44.1    | 12.9         | 16.1          | 0.9       | 10.1     | 25.9         |
> | Reject Sampling| 54.3    | 21.0         | 23.1          | 5.6       | 31.6     | 26.9         |
> |BRiTE| **76.9**    | **40.6**         | **37.0**          | **14.4**      | **57.1**     | **29.8**        |
>
> Evaluation results show that BRiTE can surpass the RS baseline with a stable margin on these bechmarks, which demonstrates the power of BRiTE in the face of **large datasets (40k data)** and boosts the performance of the state-of-art model such as *Qwen/Qwen2.5-7B*.
>
> **Weakness on Training Cost:** Proposed approach for BRiTE-SFT involves two stages of training, while traditional SFT methods authors compare with involve 2 stage of training. Authors to not provide any theoretical or actual cost estimates required to train with BRiTE. Given this process is two-staged, I assume it would use more GPU days and - depending on how much more - might not be directly comparable with direct SFT. Please include theoretical or actual cost estimates required to train with BRiTE-SFT, and add comparison with traditional SFT you used for Table 1.
>
> **Response:** Thank you for your question. We will discuss data efficiency and computational efficiency as follows:
> - *Data Efficiency:* (i) We want to emphasize that our algorithm only requires data without human annotation, making it more efficient than supervised fine-tuning (SFT) that relies on annotated data, which can be harder to collect; (ii) compared to other methods, such as rejection sampling, which also do not require data for the thinking process, our algorithm demonstrates greater efficiency since we have controlled for the same amount of data in all our experiments.
> - *Computational Efficiency:* As mentioned earlier, our algorithm does not need data with a thinking process, so it is not fair to compare our algorithm (or other methods like rejection sampling) with SFT, which relies on human-annotated, step-by-step thinking data. Moreover, both our algorithm and rejection sampling (RS) share the same goal: generating a high-quality thinking process (through RL or RS) and then fine-tuning LLMs based on that. The generation process is the main contributor to computational costs, a factor that is unavoidable for all these methods. Specifically, the first stage of our method (rollout + RL updating) and the RS filtering process (rollout + filtering) both require approximately 30 hours. This main cost is attributed to the rollout of the Qwen2.5-7B model, generating outputs for 40.6K data points with 8 samples per point. Since RL provides more concise/shorter responses, the overall computational time (rollout + RL updating) is comparable to the RS filtering process. Conversely, the training phase (SFT for RS or $\theta$-updating in BRiTE) is significantly less demanding, requiring only 3.3 hours for 9K data points from RS, or 9 hours for 24K data points from RL, for four epochs. our method achieves significantly better performance without substantially increasing computational costs.
>
>
> **Additional Question:** How do you select the best and worse response based on the correctness of y? Given 30 CoT, k will have correct answer, and (30-k) incorrect. How do you select best of k and worst of (30-k)? If it is not mentioned in the paper, please add elaboration.
>
> **Response:** We randomly select one correct answer $A^+$ from the $k$ available correct answers and one incorrect $A^-$ answer from the $(30-k)$ incorrect answers. The pairs $\{(A^+, A^-)\}$ are then used to create a pair for DPO training. We will add this clarification in the revision.

---

### Official Review · Reviewer_afMW · 2025-03-12

**Overall Recommendation:** 3

**Summary:**

The authors propose a unified graphical model to incorporate latent thinking processes and evaluation signals. Drawing inspiration from the EM approach, they unify current post-training methods into the same framework, such as SFT, RS and DPO.

**Claims And Evidence:**

I believe the authors may have exaggerated their claims, such as in the abstract where they state that their method "can even match or exceed human-annotated data." In reality, datasets like MATH have very careless annotations, so the performance of SFT may not be as effective as LLMs generating their own data. This is a point that has been widely discussed in many papers and is not necessarily an interesting finding.

**Essential References Not Discussed:**

N/A

**Experimental Designs Or Analyses:**

The improvement in the experiments is too small, and the baseline scores are somewhat low. For instance, for Llama-3-8b-Instruct, according to the [Meta-Llama-report](https://huggingface.co/meta-llama/Meta-Llama-3-8B-Instruct), the current method only matches the performance of the 4-shot approach on MATH. Given that the method still requires SFT and RL training, the improvement is quite limited.

**Methods And Evaluation Criteria:**

Regarding the evaluation benchmark, currently only tasks like code and math, which are easier to evaluate based on final answers, have been used. However, as mentioned in Section 3.4, the method should be effective for any token sequence. It would be better to include a broader range of tasks to evaluate its general applicability.

**Other Comments Or Suggestions:**

N/A

**Other Strengths And Weaknesses:**

**Strengths:**
The paper introduces an innovative graphical model to represent the LLM post-training process.

**Weaknesses:**
I did not find enough novel approaches that significantly contribute to improving LLM reasoning abilities. Instead, the paper primarily re-expresses previous methods within a unified framework.

**Questions For Authors:**

1. Has anyone evaluated the quality of the generated data? Is there any intuition that explains why the new data is of higher quality?

**Relation To Broader Scientific Literature:**

The author mentions the recent paper in related works.

**Theoretical Claims:**

I checked it briefly and did not find any issues.

---

> ### Author Rebuttal · Authors · 2025-03-31
>
> **Comments on "Claims and Evidence":**  In reality, datasets like MATH have very careless annotations, so the performance of SFT may not be as effective as LLMs generating their own data. This is a point that has been widely discussed in many papers and is not necessarily an interesting finding.
>
> **Response:** Thank you for bringing this to our attention. If a widely used dataset like MATH still struggles with careless annotations, it strengthens our motivation for research, as obtaining data that accurately reflects a high-quality thinking process is challenging. Additionally, our algorithms demonstrate their effectiveness by comparing with other algorithms, showcasing their ability to learn without relying on high-quality step-by-step data.
>
>
> **Comments on "Methods And Evaluation Criteria":** It would be better to include a broader range of tasks to evaluate its general applicability.
>
> **Response:** We perform BRiTE on the model *Qwen/Qwen2.5-7B* with a mixed dataset (the size is around 40K) of *RUC-AIBOX/STILL-3-Preview-RL-Data*, MATH, and historical AIME problems (excluding AIME 2024).  We evaluate models trained using RS baseline and BRiTE on a diverse set of reasoning benchmarks, including Math-500, Minerva Math, Olympiad Bench, AIME24, AMC23, and GPQA Diamond. The pass@1 accuracy (64 sampling times for AIME24 and AMC23 and 8 sampling times for the others) for each benchmark is reported in the table below.
> |    Method           | Math500 | Minerva_math | Olympiadbench | Aime24 | Amc23 | Gpqa_diamond |
> |----------------|---------|--------------|---------------|-----------|----------|--------------|
> | Base Model          | 44.1    | 12.9         | 16.1          | 0.9       | 10.1     | 25.9         |
> | Reject Sampling| 54.3    | 21.0         | 23.1          | 5.6       | 31.6     | 26.9         |
> |BRiTE| **76.9**    | **40.6**         | **37.0**          | **14.4**      |**57.1**     | **29.8**        |
>
> Evaluation results show that BRiTE can surpass the RS baseline with a stable margin on these bechmarks, which demonstrates the power of BRiTE in the face of **large datasets (40k data)** and boosts the performance of the state-of-art model such as *Qwen/Qwen2.5-7B*.
>
> **Weaknesses:** I did not find enough novel approaches that significantly contribute to improving LLM reasoning abilities. Instead, the paper primarily re-expresses previous methods within a unified framework.
>
> **Response:** Thank you for recognizing that our proposed framework is general, encompassing many previous methods as special cases. We also want to highligh our four contributions:
> 1. **Novel Framework and Algorithm:** The unified framework itself, which innovatively utilizes a graphical model to incorporate both thought $z$ and the optimality $o$ in the explanation of LLM reasoning, is an important contribution to theoretically characterize LLM reasoning. Furthermore, the two-stage optimization algorithm BRiTE, derived from this framework, includes various RL paradigms as special cases and introduces new learning algorithms, demonstrating the framework's utility.
> 2. **Unified Theoretical Guarantee:** We provide a unified theoretical guarantee for BRiTE, thereby offering a theoretical foundation for a broad range of algorithms (for the first time) within this framework.
> 3. **Superior Empirical Performance:** Our method significantly outperforms previous methods on bootstrapping thinking process (please see the results in our paper and the updated results in the table below).
> 4. **RL-Guided Reasoning Paradigm:** Our work demonstrates that RL-guided reasoning paths are effective for generating high-quality thinking processes. This aligns with the concurrent DeepSeek-R1 project, which also employs RL training to generate SFT data.
>
> **Additional Question:** Has anyone evaluated the quality of the generated data? Is there any intuition that explains why the new data is of higher quality?
>
> **Response:**
> We evaluated the perplexity (the mean negative log probability of the model on the responses) of the base model (Gemma-2-9b-it) using the responses from both the human-annotated data in the corresponding dataset and RL-guided data (generated by model after $\psi$-update). The results demonstrate that the RL-guided data exhibits significantly lower perplexity, which means models are easier to learn the reasoning path from the RL-guided data.
>
> | Perplexity             |GSM8K | MATH |
> |----------------------|------------|------------|
> | Human-annotated data | 4.14       | 2.84       |
> | RL-guided data  | 1.64       | 1.40       |
>
> Due to the page limit of the rebuttal, we could not provide a concrete comparison between the RL-guided response and the human-annotated response, which shows that the RL-guided reasoning path can provide more detailed and clear steps for thinking and is easier for the model to learn and derive the correct answer. If you are interested, we are pleased to provide this comparison in the new round of rebuttal.

---

> > ### Comment · Reviewer_afMW · 2025-04-08
> >
> > Thank you for the detailed response. The new experiments addressed most of my concerns. I have raised my score.

---

### Official Review · Reviewer_UuED · 2025-03-25

**Overall Recommendation:** 3

**Summary:**

This paper works on the reasoning process generation problem. The authors propose formalizing the reasoning problems as a probabilistic graphical model involving input, latent rationals, answer, and evaluation signal. Then they presented the BRiTE algorithm with a theoretical guarantee on convergence. The proposed method outperforms rejection sampling and SFT on math and coding benchmarks.

**Claims And Evidence:**

Mostly yes.
1. The authors provided a detailed explanation of their reasoning framework and proved the theoretical guarantee.
2. They mainly experimented with four models on GSM8K and MATH. BRiTE is consistently better than rejection sampling and is only slightly worse than SFT when using the Mistral model.
3. In the results, when the deepseek model was used on coding generation tasks. BRiTE-SFT shows better performance with only one exception.

**Essential References Not Discussed:**

not found

**Experimental Designs Or Analyses:**

The paper compared their algorithm with rejection sampling, SFT, and iterative DPO. I think the experiment is lightweight but enough for claims. But definitely more datasets and benchmarks would increase the soundness.

**Methods And Evaluation Criteria:**

Yes, the two benchmarks are widely used, and the evaluation metrics are reasonable.

**Other Comments Or Suggestions:**

no

**Other Strengths And Weaknesses:**

Strengths:
1. Originality: new generalizable reasoning frame, method, and theoretical analysis.
2. Presentation is clear with minor typos

Weakness:
1. Reasoning task diversity is limited
2. BRiTE could introduce heavy computational overhead, but the performance improvement is not significantly better than other efficient methods.

**Questions For Authors:**

Hi,
1. Could the authors discuss the algorithm in terms of efficiency? Including data efficiency and computation efficiency.
2. It would be very helpful if the authors could explain the algorithm in steps. In my current understanding. The main contribution is to provide a new loss to optimize. Please let me know if my understanding is incorrect.

**Relation To Broader Scientific Literature:**

The proposed framework incorporated the popular CoT idea, rejection sampling. And can be applied in RLHF and SFT scenarios.

**Theoretical Claims:**

I went over the proofs of convergence guarantee briefly. I did not find any issues.

---

> ### Author Rebuttal · Authors · 2025-03-31
>
> **Weakness:**
> (1) Reasoning task diversity is limited. (2) The performance of BRiTE is not significantly better than other efficient methods.
>
> **Response:** We perform BRiTE on the model *Qwen/Qwen2.5-7B* with a mixed dataset (the size is around 40K) of *RUC-AIBOX/STILL-3-Preview-RL-Data*, MATH, and historical AIME problems (excluding AIME 2024).  We evaluate models trained using RS baseline and BRiTE on a diverse set of reasoning benchmarks, including Math-500, Minerva Math, Olympiad Bench, AIME24, AMC23, and GPQA Diamond. The pass@1 accuracy (64 sampling times for AIME24 and AMC23 and 8 sampling times for the others) for each benchmark is reported in the table below.
> |    Method           | Math500 | Minerva_math | Olympiadbench | Aime24 | Amc23 | Gpqa_diamond |
> |----------------|---------|--------------|---------------|-----------|----------|--------------|
> | Base Model          | 44.1    | 12.9         | 16.1          | 0.9       | 10.1     | 25.9         |
> | Reject Sampling| 54.3    | 21.0         | 23.1          | 5.6       | 31.6     | 26.9         |
> |BRiTE| **76.9**    | **40.6**         | **37.0**          | **14.4**      | **57.1**     | **29.8**        |
>
> Evaluation results show that BRiTE can surpass the RS baseline with a stable margin on these bechmarks, which demonstrates the power of BRiTE in the face of **large datasets (40k data)** and boosts the performance of the state-of-art model such as *Qwen/Qwen2.5-7B*.
> **Question 1:**
> Could the authors discuss the algorithm in terms of efficiency?
>
> **Response:** Thank you for your question. We will discuss data efficiency and computational efficiency as follows:
> - *Data Efficiency:* (i) We want to emphasize that our algorithm only requires data without human annotation, making it more efficient than supervised fine-tuning (SFT) that relies on annotated data, which can be harder to collect; (ii) compared to other methods, such as rejection sampling, which also do not require data for the thinking process, our algorithm demonstrates greater efficiency since we have controlled for the same amount of data in all our experiments.
> - *Computational Efficiency:* As mentioned earlier, our algorithm does not need data with a thinking process, so it is not fair to compare our algorithm (or other methods like rejection sampling) with SFT, which relies on human-annotated, step-by-step thinking data. Moreover, both our algorithm and rejection sampling (RS) share the same goal: generating a high-quality thinking process (through RL or RS) and then fine-tuning LLMs based on that. The generation process is the main contributor to computational costs, a factor that is unavoidable for all these methods. Specifically, the first stage of our method (rollout + RL updating) and the RS filtering process (rollout + filtering) both require approximately 30 hours. This main cost is attributed to the rollout of the Qwen2.5-7B model, generating outputs for 40.6K data points with 8 samples per point. Since RL provides more concise/shorter responses, the overall computational time (rollout + RL updating) is comparable to the RS filtering process. Conversely, the training phase (SFT for RS or $\theta$-updating in BRiTE) is significantly less demanding, requiring only 3.3 hours for 9K data points from RS, or 9 hours for 24K data points from RL, for four epochs. our method achieves significantly better performance without substantially increasing computational costs.
>
>
> **Question 2:**
> It would be very helpful if the authors could explain the algorithm in steps. In my current understanding. The main contribution is to provide a new loss to optimize.
>
> **Response:** Regarding the implemented algorithm, the main contribution is providing a new approach to generating high-quality latent thinking processes through RL objective with a reward function (as detailed in Section 3.4). Specifically, inspired by our theory, the learning algorithm guarantees provable convergence by iteratively updating equations (3.4) and (3.5). Equation (3.5) is relatively easier to implement, while the key challenge lies in solving equation (3.4). Guided by Proposition 3.7, solving the intractable distribution in (3.4) is equivalent to addressing an entropy-regularized Markov Decision Process (MDP) with the reward function $\log P(z, y, o\mid x)$. This can be further specified using the graphical model; for instance, if we let $P(o = 1 | z, y, x) = \exp(\beta \cdot R(x, z, y))$, we can define the reward function as $\log P(z, y | x) + \beta \cdot R(x, z, y)$ with a tuning parameter $\beta$. This can be attributed to each token through the autoregressive policy property, allowing us to assign the reward to the last token.
>
> Beyond the algorithmic contributions, our main contribution includes a **new unified framework, theoretical guarantee, superior performance, and the demonstration of RL-guided reasoning's potential**. Due to space constraints, we refer you to our detailed response to Reviewer afMW.

---

### Decision · Program_Chairs · 2025-05-01

**Decision:**

Accept (poster)

**Comment:**

**summary**:
This paper introduces BRiTE, a framework for reasoning process generation that formalizes latent rationales using a probabilistic model and optimizes them through reinforcement learning without needing labeled chain-of-thought data. The method is theoretically grounded and shows consistent, albeit modest, improvements across math and code reasoning tasks with various language models.

**Pros**
1. Proposes a novel and theoretically sound framework incorporating latent rationales and evaluation signals.
2. Demonstrates consistent improvements across multiple models on math and code generation tasks.
3. Does not require human-annotated rationales, making it more scalable for training.

**Cons**
1. Experimental scope is limited to math and code tasks, with no evaluation on broader reasoning benchmarks.
2. Performance gains over strong baselines are relatively small.
3. BRiTE involves a multi-stage training process that may introduce significant computational overhead without clear cost analysis.